# Plasmacytoid dendritic cells have divergent effects on HIV infection of initial target cells and induce a pro-retention phenotype

Orion Tong[1,2], Gabriel Duette[1], Thomas R. O'Neil[1,2], Caroline M. Royle[1], Hafsa Rana[1,2], Blake Johnson[1,2], Nicole Popovic[1,2], Suat Dervish[3], Michelle A. E. Brouwer[1,4], Heeva Baharlou[1,2], Ellis Patrick[1,5], Grahame Ctercteko[1,6†], Sarah Palmer[1,2], Eunok Lee[1], Eric Hunter[7], Andrew N. Harman[1,8], Anthony L. Cunningham[1,2]*, Najla Nasr[1,8]*

1 Centre for Virus Research, The Westmead Institute for Medical Research, Westmead, New South Wales, Australia, 2 The University of Sydney, Faculty of Medicine and Health, Sydney, New South Wales, Australia, 3 Westmead research Hub, The Westmead Institute for Medical Research, Westmead, New South Wales, Australia, 4 Department of Internal Medicine, Radboud Centre for Infectious Diseases, Radboud Institute of Molecular Life Sciences, Radboud University Medical Centre, Nijmegen, The Netherlands, 5 The University of Sydney, School of Mathematics and Statistics, Faculty of Science, Sydney, New South Wales, Australia, 6 Westmead Hospital, Westmead, New South Wales, Australia, 7 Emory Vaccine Centre, Atlanta, Georgia, United States of America, 8 The University of Sydney, School of Medical Sciences, Faculty of Medicine and Health, Sydney, New South Wales, Australia

† Deceased.

* tony.cunningham@sydney.edu.au (ALC); najla.nasr@sydney.edu.au (NN)

**Data Availability Statement:** All relevant data are within the manuscript and its Supporting Information files.

## Abstract

Although HIV infection inhibits interferon responses in its target cells *in vitro*, interferon signatures can be detected *in vivo* soon after sexual transmission, mainly attributed to plasmacytoid dendritic cells (pDCs). In this study, we examined the physiological contributions of pDCs to early HIV acquisition using coculture models of pDCs with myeloid DCs, macrophages and the resting central, transitional and effector memory CD4 T cell subsets. pDCs impacted infection in a cell-specific manner. In myeloid cells, HIV infection was decreased via antiviral effects, cell maturation and downregulation of CCR5 expression. In contrast, in resting memory CD4 T cells, pDCs induced a subset-specific increase in intracellular HIV p24 protein expression without any activation or increase in CCR5 expression, as measured by flow cytometry. This increase was due to reactivation rather than enhanced viral spread, as blocking HIV entry via CCR5 did not alter the increased intracellular p24 expression. Furthermore, the load and proportion of cells expressing HIV DNA were restricted in the presence of pDCs while reverse transcriptase and p24 ELISA assays showed no increase in particle associated reverse transcriptase or extracellular p24 production. In addition, pDCs also markedly induced the expression of CD69 on infected CD4 T cells and other markers of CD4 T cell tissue retention. These phenotypic changes showed marked parallels with resident memory CD4 T cells isolated from anogenital tissue using enzymatic digestion. Production of IFNα by pDCs was the main driving factor for all these results. Thus, pDCs may reduce HIV spread during initial mucosal acquisition by inhibiting replication in myeloid cells while reactivating latent virus in resting memory CD4 T cells and retaining them for immune clearance.

**Funding:** This work was funded by the National Health and Medical Research Centre (NHMRC) via Project Grant #APP1106442 (ALC, NN, AH) and Investigator Grant APP1177942 (ALC). TRO'N and HR are supported by research training program at the University of Sydney. The funders played no role in the study design, data collection and analysis, decision to publish, or preparation of the manuscript.

**Competing interests:** The authors have declared that no competing interests exist. Author Dr Grahame Ctercteko was unable to confirm their authorship contributions. On their behalf, the corresponding author has reported their contributions to the best of their knowledge.

## Author summary

IFNs constitute one of the first and most important innate immune controls to restrict initial viral replication and spread. As HIV has evolved mechanisms to block IFN-I induction in its target cells, but not in infiltrating pDCs, understanding how pDCs influence HIV infection of target cells upon initial transmission is critical to prevent or control initial infection. Therefore, we modelled the early events occurring immediately as HIV enters the human genital mucosa. We showed that IFNα secreting pDC compensated for HIV inhibition of IFN-I production in its target cells in two different ways: i) reduced infection in DCs and macrophages which would limit viral spread to resident or newly infiltrating memory CD4 T cells; ii) reactivation of latent HIV in all subsets of resting memory CD4 T cell subsets, accompanied by limited viral spread, upregulation of MHC-I and induction of a tissue retention phenotype. The increased HIV protein, MHC-I expression and retention may enhance exposure to CD8 T cell surveillance. This model suggests that IFNα reactivation of latent HIV combined with adoptive immunotherapy using CD8 T cells or those expressing chimeric antigen receptors (CAR) could provide a novel 'kick and kill' approach to eradicate HIV reservoirs.

## Introduction

Type 1 interferon (IFN-I; α, β, ε, ω and κ) constitute one of the first and most important innate immune control mechanisms to prevent initial viral infection. IFNα and β are secreted by immune cells in response to pathogens to induce an antiviral state in surrounding cells [1]. We have shown that HIV evades immune recognition by inhibiting IFNβ production in monocyte derived dendritic cells (MDDC) and macrophages (MDM) [2–4]. However, these cells respond directly to HIV by expressing IFN stimulated genes (ISG) [3,5], which limit HIV replication and cytopathicity. A distinct but comparable mechanism of IFN-I inhibition also occurs upon infection of CD4 T cells [6,7]. Despite IFNβ inhibition, IFNα is detected in the circulation of HIV patients within 1–2 weeks [8] with plasmacytoid dendritic cells (pDCs) likely being the main source [9]. Upon initial cervical simian immunodeficiency virus (SIV) infection of macaques, pDCs were shown to infiltrate in cervix within 1–2 days post infection (dpi) and secrete large amounts of IFNα, followed by recruitment of circulating blood effector CD4 T cells to this site [10], which is also likely populated with HIV infected and migrating myeloid DCs, macrophages and tissue resident memory (TRM) CD4 T cells.

Whilst CD4 T cells are well established to be prime target cells for HIV infection, dendritic cells (DC) and macrophages are among the first cells in the anogenital mucosa to encounter HIV during sexual transmission [11]. They are susceptible to viral entry and infection, and influence HIV dissemination through viral transfer to T cells both at local sites and systematically. We have shown that human tissue derived epidermal CD11c$^+$ DC and Langerhans cells (LC) can both efficiently transfer HIV to its main target, CD4 T cells [11,12]. CD4 T TRM cells are believed to be the main CD4 T cells present in peripheral tissues [13]. Little is known on whether infiltrating CD4 T cell subsets become TRM in anogenital tissues and what regulates their retention and survival after pathogen clearance. Recent studies have identified CD69 and, to a much lesser extent, PD1 and CD103 as markers delineating CD4 TRM from circulating memory T cells across multiple mucosal and lymphoid tissues [14,15]. These cells can also be identified by the expression of core gene signatures including chemokine homing markers (CXCR6, CX3CR1), long term survival receptors (CD127) and the downregulation of CXCR3, CD62L and CCR7 receptors to avoid exit from tissues [14]. Recently, CD4 TRM T cells

extracted from the cervical mucosa of HIV patients were shown to harbour up to 200 times more provirus per cell than their circulating counterparts isolated from blood [16]. In the periphery, upon antigen presentation by DCs, naïve CD4 T cells become activated and differentiate into specialised subsets of T cells which migrate to sites of pathogen exposure to clear infection. A minority of these T cells will then differentiate into long-lived memory cells: central memory (TCM), transitional memory (TTM) or effector memory (TEM) depending on the strength of stimulation, with stronger signals leading to TEM formation [17]. Whilst HIV productively replicates in activated CD4 T cells, contributing to their depletion and progression to AIDS, latent infection is established in resting memory CD4 T cells and can result in the development of HIV viral reservoirs. These reservoirs may persist for years and can rapidly reactivate to propagate HIV replication when antiretroviral therapy (ART) is discontinued [18].

To date, the role of pDCs in early SIV/HIV infection have been inferred from simian studies [19] and humanised mice [20]. In humans, pDC and IFN-based strategies have previously been investigated mainly as complements to latency-reversing agents to eradicate the latent reservoir in patients being treated with ART [21], but little work has been conducted on their effects during the establishment of infection. Therefore, in this study, we have developed a cell culture model of HIV-infected MDDCs, MDMs and resting memory CD4 T cells whereby bona-fide pDCs are added to cultures at 2 dpi, followed by monitoring of ensuing changes in HIV infection and expression of various cellular markers. This allowed us to elucidate the interactions between pDCs and HIV infected target cells *in vitro* to determine how pDCs may help restrict the spread of HIV from the site of initial anogenital infection, partially substituting for the lack of endogenous IFN. Our data indicates that via IFNα production, pDCs inhibit HIV replication in MDDCs and MDMs but instead enhance expression of intracellular HIV proteins in resting blood CD4 memory T cells. Rather than resulting from increased *de novo* infection, this could be mainly attributed to the reactivation of latently infected cells and was accompanied by the lack of a corresponding increase in extracellular HIV and diminished HIV spread or expansion of intracellular integrated HIV DNA. Alongside increased HIV p24 expression, pDCs also induced expression of the tissue retention marker CD69 and major histocompatibility class (MHC)-I. Taken together this suggests that pDCs restrict local and systemic HIV spread *in vivo* and may enhance the early detection and elimination of latently HIV-infected CD4 T cells by the immune system.

## Results

### pDCs decreased HIV infection in MDDCs and MDMs

To examine the effects pDCs have during initial HIV infection, MDDCs and MDMs infected with HIV$_{BaL}$ for 48 hours were cultured alone or with pDCs for a further 72 hours (Fig 1A). HIV infection was assessed 5 dpi by intracellular p24 staining. Coculture with pDCs significantly reduced the percentage of HIV p24+ cells in MDDCs and MDMs (Fig 1B and 1C) and the number of virions per cell (Fig 1D). No p24+ pDCs were detected with MOIs ranging from 0.5–6. This was similar to a recent study showing that when authentic pDCs are separated from Axl+ Siglec6+ myeloid DC (ASDCs), they cannot be infected with HIV and cannot transfer virus to CD4 T cells [22]. In summary, productive HIV infection and spread in MDDCs and MDMs was markedly and significantly reduced by pDCs.

### Mechanisms of pDC mediated HIV inhibition in MDDCs and MDMs

To understand how pDCs decreased HIV infection and spread in MDDCs and MDMs, we investigated the IFN-induced ISGs, the maturation status of cells and CCR5 expression levels.

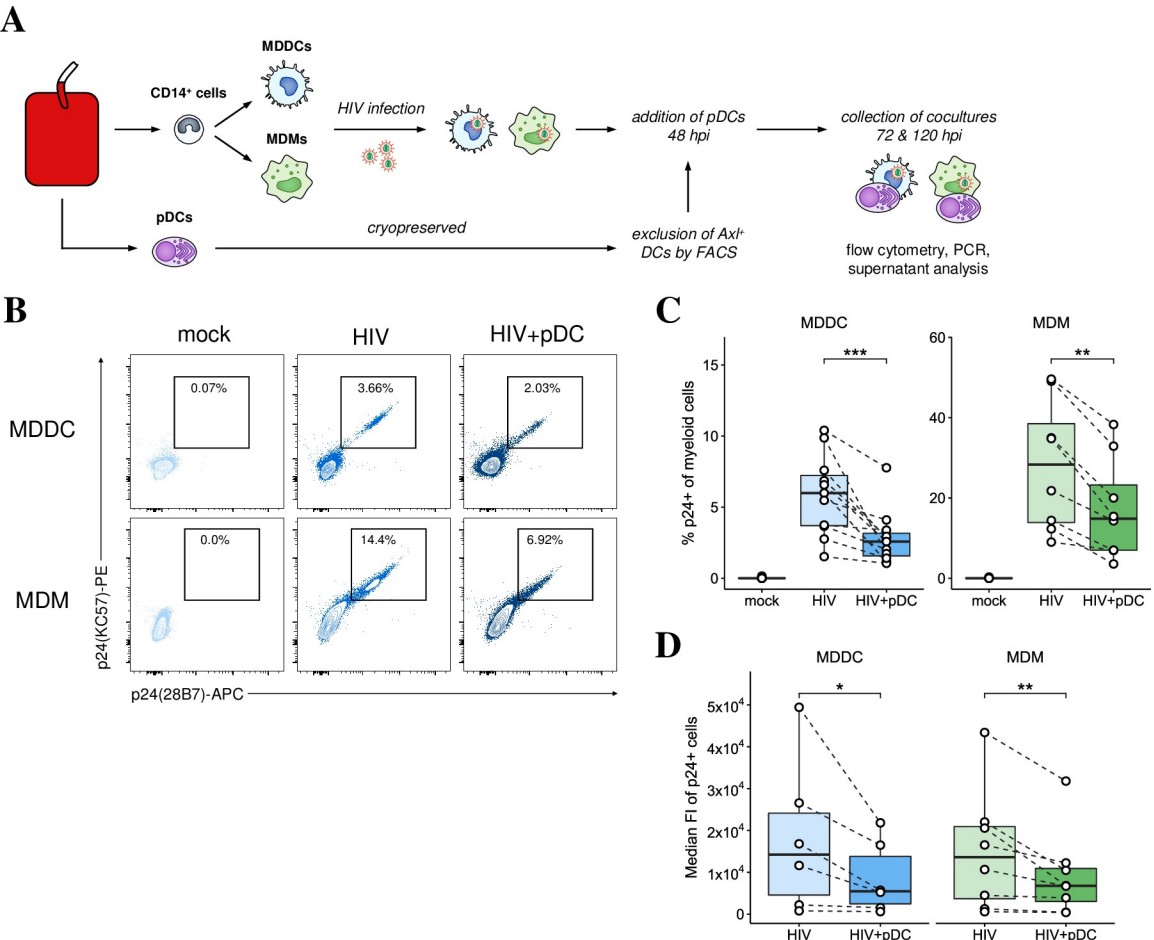

**Fig 1. Reduced HIV infection in MDDCs and MDMs upon addition pDCs.** (A) Workflow for generating MDDC-pDC and MDM-pDC cocultures: CD14+ cells were isolated from blood and differentiated for 5 days into MDDCs and MDMs. They were infected with HIV$_{BaL}$ (MOI of 0.75) overnight, washed and then cultured for further 2 days prior to the addition of pDCs. (B) Representative dot plots showing p24 expression at 5 dpi in MDDCs and MDMs that were either mock treated, infected with HIV, or infected and cocultured with pDCs. (C) Mean percentage of p24+ MDDCs (n = 11) and MDMs (n = 8). (D) Median fluorescent intensity (FI) of p24+ MDDCs (n = 6) and MDMs (n = 8). For all data, *p < 0.05, **p < 0.01, ***p < 0.001 by Wilcoxon signed rank test.

On day 3 pi, selected antiviral ISGs known to inhibit HIV replication, *IFI44L*, *IFIT1-3*, *IFITM1-3*, *OAS1* and *Viperin*, were induced at low levels in HIV infected MDDCs and MDMs in the absence of both pDCs and IFN as we have previously reported [2,3,5] (S1A Fig). However, upon addition of pDCs to HIV-infected cultures, *IFN* mRNA and ISG expression were induced at significantly higher levels in cocultured MDDC-pDCs (S1A Fig) and MDM-pDCs. We controlled for this expression in cocultures by examining what ISGs were upregulated when pDCs alone were exposed to a cell-free HIV inoculum (S1B Fig). Although IFNα/β and λ1 were induced, none of the ISGs that were significantly induced in cocultures were upregulated in pDCs alone. This strongly suggests that the ISGs detected in cocultures were primarily expressed in MDDCs and MDMs while pDCs were the main source of types I and III IFN induction. Furthermore, when we performed RNA sequencing on pDCs infected with HIV for 18 hours, we detected the 13 subtypes of *IFN-I* (α, β, ε, ω, κ), *IFN-III* (*IFNλ1–4*) (Fig 2A) and *TNFα* (S1C Fig). The production of IFNα and β was also detected by ELISA in supernatants derived from cocultures (S1D Fig) and TNFα has been previously detected in supernatants of HIV-infected pDCs [23,24].

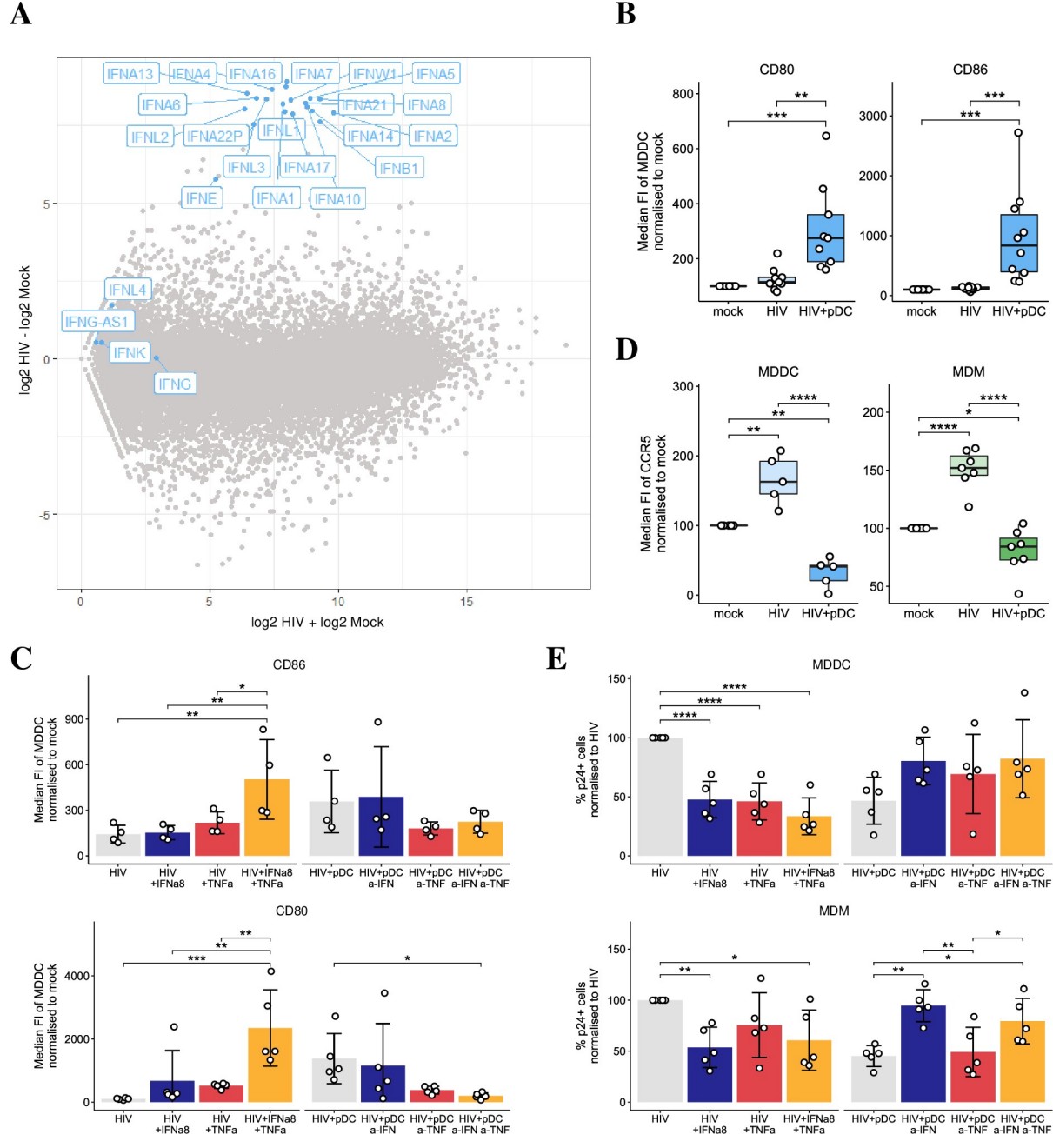

**Fig 2. Mechanisms of reduced infectivity in MDDCs and MDMs.** (A) RNAseq data showing IFN-subtype gene induction in pDCs at 18 hours post HIV exposure versus mock. (B-C) The median FI of maturation markers, CD80 (n = 9) and CD86 (n = 10) in MDDCs that were either mock, HIV infected MDDCs in the presence and absence of pDCs, and in HIV infected MDDCs treated with exogenous rIFNα8 and/or rTNFα, HIV infected MDDCs treated with antibodies to blocking IFN and/or TNF signaling in pDC cocultures (n = 4 for CD80, n = 5 for CD86). Data is shown as normalized to 100% in mock infected. (D) pDC decreased CCR5 median FI in HIV-infected MDDCs (n = 5) and MDMs (n = 7). Data is shown as normalized to 100% in mock infected. (E) Changes in p24 expression at 5 dpi in MDDCs (n = 5) (top panel) and MDMs (n = 5) (bottom panel) upon addition of exogenous rIFNα8 and/or rTNFα, or upon blocking IFN and/or TNF signaling in pDC cocultures. Data was normalized to 100% in HIV infected cells in the absence of pDCs. For all data, $^*p < 0.05$, $^{**}p < 0.01$, $^{***}p < 0.001$, $^{****}p < 0.0001$ by repeated measures ANOVA with Tukey post-hoc test.

As mature MDDCs are known to be resistant to productive HIV infection [25], we assessed whether the maturation markers CD80 and CD86 are increased in MDDCs cocultured with pDCs or treated with both recombinant (r) IFNα8 and/or rTNFα. The choice of rIFNα8 was based on data from *in vitro* and humanized mice studies showing IFNα8 and IFNα14 are more efficient at inhibiting HIV viral replication [26,27] due to their higher affinities for the IFN receptor (IFNAR) and increased downstream expression of antiviral proteins. CD80 and CD86 expression were significantly upregulated to higher levels on MDDCs (Fig 2B) and MDMs (S2A Fig) upon coculture with pDCs and similarly, upon addition of both exogenous rIFNα8 and rTNFα with an additive effect compared to each treatment alone (Fig 2C, left panel and S2B Fig). To validate these results, we inhibited the IFN and TNF signaling pathways by blocking the IFNAR and/or TNFR1 receptors and treating with neutralising antibodies for IFNα/β and/or TNFα for 2 hours prior to co-culture with pDCs. Antibody inhibition of both IFN-I and TNF signalling significantly downregulated CD86 to a much greater level compared to each individual treatment in MDDCs (Fig 2C, right panel) and MDMs (S2B Fig) while CD80 was significantly downregulated in MDMs and showed a trend towards downregulation with the combined treatment in MDDCs. Further, as mature MDDCs downregulate CCR5, we assessed CCR5 levels and detected a significant decrease of its expression on both MDDCs and MDMs upon coculture with pDCs (Fig 2D).

To confirm that production of IFN-I and/ or TNFα by pDCs was responsible for the decreased p24 expression in MDDCs and MDMs, cells were treated with exogenous rIFNα8 and/or rTNFα. As observed in cocultures with pDCs, there was a significant decrease in the percentage of p24$^+$ cells in both MDDCs and MDMs treated with rIFNα8, while rTNFα reduced p24$^+$ cells only in MDDCs but not MDMs (Fig 2E). In addition, neutralising IFN-I and/or TNFα signalling in cocultures increased p24 expression in MDDCs while only neutralising IFN-I increased p24 expression in MDMs (Fig 2E). This indicates that the decrease in p24 was partially mediated by both IFN-I and TNFα in MDDCs and only via IFN-I in MDMs. In summary, restriction of HIV replication in MDDCs and MDMs cocultured with pDCs is likely to be mediated via IFN production, antiviral ISG induction, CCR5 downregulation and maturation, which all collectively prevent HIV viral spread.

### pDCs increase intracellular p24 expression but not extracellular virus production in resting memory CD4 T cell in a cell subset-dependent manner

To assess HIV infection of resting memory T cells, TCM, TTM and TEM were either mock or HIV$_{BaL}$ infected in the presence and absence of pDCs followed by intracellular p24 staining at 5 dpi (Fig 3A). The expression of p24 upon infection ranged from 0.2–2% to 2.1–12% across the subsets, being the lowest in TCM followed by an increase from TTM to TEM (Fig 3B and 3C) and there was no change in the number of virions per cell (S3A Fig). There was also an increased expression of CCR5 levels from TCM to TEM (Fig 3D), consistent with the patterns of p24 expression across the subsets. Upon coculture with pDCs (1 pDC: 3 CD4 T cells), CCR5 expression did not change but a significant increase in the percentage of p24$^+$ cells was observed across all subsets (Fig 3C). When resting memory CD4 T cell subsets were infected with a CCR5-utilising primary HIV transmitted/founder (HIV$_{TF}$) strain, they also showed significant and progressive increased p24 expression from CM to TM and EM similar to HIV$_{BaL}$ infection, albeit at lower overall levels (Fig 3E). The expression of p24 was similarly significantly increased upon addition of pDCs. However, when we assessed extracellular virion production via a reverse transcriptase assay and p24 ELISA, surprisingly both assays showed that there was no increase in viral release in supernatants derived from cocultures (Fig 3F and 3G).

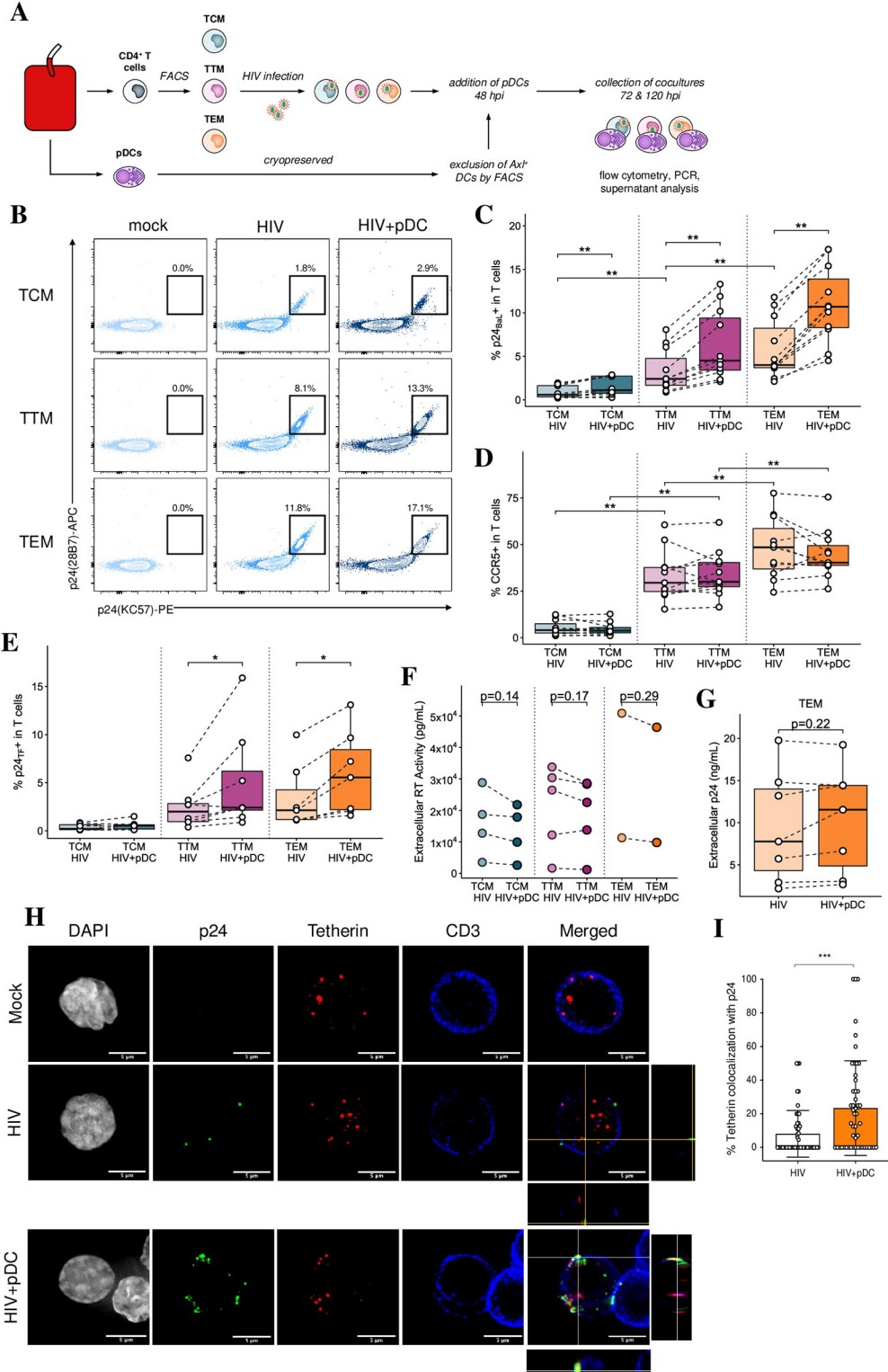

**Fig 3. Addition of pDCs increased intracellular HIV expression but not extracellular virus production in memory CD4 T cells.** (A) Workflow for generating memory T cell-pDC cocultures: memory CD4 T cells were sorted into resting TCM, TTM and TEM subsets. Sorted cells were infected with either HIV$_{BaL}$ or HIV$_{TF}$ (MOI of 0.75) overnight. They were cultured for further 2 days before the addition of pDCs. (B) Representative dot plots showing p24 expression at 5 dpi in resting TCM, TTM and TEM that were infected with HIV$_{BaL}$ in the absence (HIV) or presence

of pDCs (HIV+pDC). (C) Mean percentage of p24$^+$ cells at 5 dpi with HIV$_{BaL}$ (n = 11). (D) CCR5 expression across T cell subsets in the presence and absence of pDCs (n = 11). (E) Mean percentage of p24$^+$ cells at 5 dpi with the HIV$_{TF}$ in the presence or absence of pDCs (n = 7). (F) Assessment of extracellular viral production by the reverse transcriptase assay using supernatants derived from infected resting CD4 T cell subsets in the absence (HIV) or presence of pDCs (HIV+pDC) (n = 2–5). (G) Same as in F but using p24 ELISA on supernatants derived from infected resting TEM cells (n = 2). For all data, *p < 0.05, **p < 0.01 by Wilcoxon signed-rank test with multiple testing correction (C, D, E) or paired two-sided t-test (F, G). (H-I) Co-localisation of p24 and Tetherin in TEM by immunofluorescent microscopy using anti-DAPI (grey), anti-p24 (green), anti-Tetherin (red) and anti-CD3 (blue). Representative multiple Z-plane images were acquired to provide a merged maximum intensity projection with an orthogonal view to visualise colocalization of Tetherin and p24 (H). Bar graph (I) showing % of Tetherin co-localisation with p24+ cells (n = 50 cells, unpaired T test two tailed, ***p < 0.001).

This indicates that the increase in intracellular p24 expression in cocultures did not result in an increase in extracellular virus production. Furthermore, when we assessed supernatant infectivity via infection of TZMbl reporter cells, we found a significant decrease in the number of infected TZMbl cells upon their inoculation with supernatants derived from the cocultures (S3B Fig). This is consistent with two effects; firstly an effect of IFN inducing late acting ISGs such as Tetherin. Upon investigation by Imagestream flow cytometry, we found a baseline level of Tetherin expression in mock infected memory T cells which was similar in HIV infected cells but was markedly and significantly increased upon the addition of pDCs to these infected cells (S3C and S3D Fig). Immunofluorescent microscopy showed wide distribution of Tetherin in the CD4 T cell cytoplasm with sub-cell membranous foci with all treatments. However, with the addition of pDCs to HIV infected cells, there was a marked and significant increase in colocalization between p24 and Tetherin of 24% compared to only 8% in HIV infected cells in the absence of pDCs. This colocalization was mostly at the cell surface, and thus likely to reduce the release of budded HIV-1 virions from infected cells (Fig 3H and 3I).

Another intracellular antiviral mechanism is APOBEC3G mediated G to A hypermutation which causes stop codons in HIV-DNA during reverse transcription preventing further HIV replication and infection [28,29]. To determine if the presence of pDC enhances APOBEC3G hypermutation of HIV, we performed single-proviral sequencing to obtain individual HIV-DNA *env* V1-V3 sequences from TTM and TEM infected with HIV-1 BaL and cocultured with either pDC or the supernatant of pDC. We found no evidence for G to A hypermutation within the viral sequences isolated from the TTM and TEM cells which were infected with HIV and then cocultured with either pDC or the supernatant of pDC. The viral sequences containing premature stop codons were rare within the memory T-cells alone or those cocultured with pDC or the supernatant of pDC (S4A Fig). In addition, the frequency of G to A mutations within the viral sequences was significantly lower than the non-G to A mutations across all experimental conditions (S4B–S4D Fig) with p-value of <0.0001–0.037. Therefore, our sequencing analysis shows little evidence for the APOBEC3G mediated hypermutation elicited by pDCs, suggesting little effect in exiting virus. In summary, there was restriction of viral egress from HIV infected cells when cocultured with pDCs, despite enhanced cytoplasmic p24 expression, probably partly by Tetherin. The second effect was a further decrease in extracellular virus infectivity and spread probably by soluble factors, especially type I IFN, secreted by pDCs.

## Mechanism of pDC enhancement of intracellular HIV p24 in resting memory CD4 T cells

To ensure that the increased p24 expression in memory CD4 T cells cocultured with pDCs was not due to a defect in IFN production and/or the induction of ISGs in cocultures, the gene expression of type I and III IFNs and antiviral ISGs was assessed by RT-qPCR. Coculture of

pDCs with infected TCMs, TTMs and TEMs resulted in significant induction of *IFN* by pDCs (S5A Fig) and ISG gene induction in all CD4 T cell subsets (S5B and S5C Fig). However, no *IFN* or ISG induction was detected in HIV infected resting memory CD4 T cell subsets in the absence of pDCs. Thus, coculture of pDCs with infected memory CD4 T cells resulted in significant IFN and ISG induction.

We next investigated whether T cell activation by pDCs might explain the increase in intracellular p24 expression by assessing the expression of early (CD69), intermediate (CD25) and late (CD38 and HLA-DR) activation markers on mock-infected and HIV-infected TCM, TTM and TEM in the presence and absence of pDCs (S5D–S5F Fig). There was an increase in CD25 (0.6–2.7%), CD38 (2.1–20%) and HLA-DR (0.1–8.1%) in most T cell subsets in the presence of pDCs. This increase was small (<10%) but significant in the proportions of TEM expressing CD25 and HLADR. However, pDCs markedly and significantly induced the expression of CD69, in a subset-specific manner. Induction on TCM was minor (3.6–27%) but increased progressively from TTM to TEM (14–50% and 18–69%, respectively) (Fig 4A and 4B). When cellular activation was assessed 10 days pi., CD4 T cell subsets still predominantly expressed only CD69, with very low CD25 and CD38 expression. Levels of p24 expression and CD69 induction were significantly correlated across all subsets with individuals expressing the highest p24 in infected TEM cocultured with pDCs having the highest degree of CD69 induction (Fig 4C). CD69 was expressed to a significantly higher degree than other activation markers on the total TEM population and on p24$^+$ cells (Fig 4D). In addition, a significant enrichment of CD69 expression was detected on p24$^+$ cells compared to the total TEM population. Similar

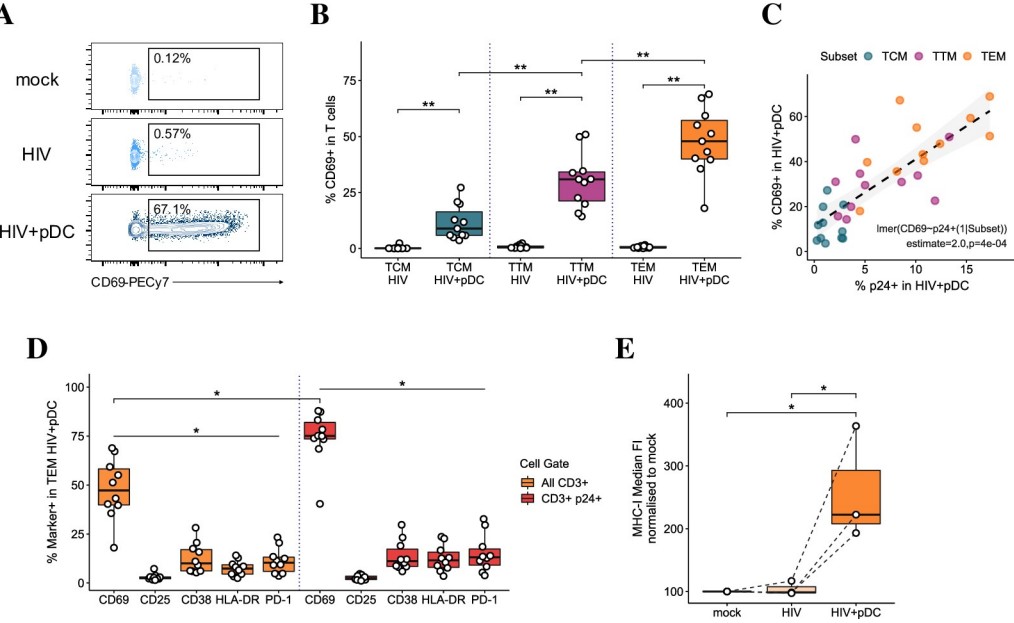

**Fig 4. Induction of CD69 in memory CD4 T cells upon addition of pDCs.** (A) Representative dot plots showing CD69 expression in resting TEM CD4 T cells at 5 dpi. (B) Data represents the mean percentage of CD69$^+$ on resting memory TCM, TTM and TEM CD4 T cell subsets. n = 11, $^{**}$p<0.01 by Wilcoxon-signed-rank test with multiple testing correction. (C) Pearson's correlation of CD69 and p24 expression across CD4 T cell subsets upon addition of pDCs (n = 11 per subset). (D) Expression of T cell activation markers (CD69, CD25, CD38, HLA-DR and PD1) in cocultures on total TEM (orange bars) and on p24$^+$ TEM (red bars). n = 11, $^*$p < 0.05 for CD69 vs all other markers indicated in total TEM and p24$^+$ TEM, and upon comparison between total TEM and p24$^+$ TEM by Wilcoxon signed-rank test with multiple testing correction. **(E)** The median FI of MHC-I expression on TEM cells in mock-infected cultures, and upon HIV infection in the absence and presence of pDCs (n = 3, $^*$p < 0.05 by repeated measures ANOVA with Tukey post-hoc test). Data is shown as normalized to 100% in mock infected.

patterns of CD69 induction and enrichment on p24$^+$ cells were detected on all resting memory CD4 T cell subsets when infected with the primary HIV$_{TF}$. Thus, pDCs selectively induced the expression of the early activation and tissue retention marker, CD69, on resting memory T cells across a prolonged timespan.

To determine whether the selective expression of CD69 and the lack of increased virus production extracellularly in cocultures were due to cell exhaustion preventing the CD4 T cells from completing a full replication cycle required for synthesis of new virions, we assessed exhaustion markers including T-cell immunoglobulin mucin-3 (Tim-3), cytotoxic T lymphocyte-associated antigen 4 (CTLA-4), T cell immunoreceptor with Ig and ITIM domains (TIGIT), lymphocyte-activation gene 3 (Lag-3) and the programmed cell death protein 1 (PD-1). TIGIT was basally expressed (~20%) on all CD4 T cell subsets and did not change in cocultures. Lag3, CTLA4 and Tim-3 were not expressed basally or induced in cocultures. However, PD-1 was induced upon the addition of pDCs to infected resting memory CD4 T cell subsets and was less than 5% in TCM and 12±6% in TTM and TEM (S5G Fig). As cellular exhaustion is best defined by expression of multiple exhaustion markers, we concluded that pDCs did not induce an exhausted phenotype but were more likely inducing the expression of tissue retention proteins PD-1 and CD69 in resting memory CD4 T cells.

Lastly, we assessed the expression of MHC-I molecules on resting memory CD4 T cells upon addition of pDCs, given the presentation of intracellular viral proteins on these molecules to CD8 T cells is required for potential viral clearance. MHC-I expression was found to be significantly upregulated in TEM in the presence of pDCs (Fig 4E), indicating pDCs may enhance the capacity for infected cells to be recognised by the immune system.

## Mechanisms of p24 and CD69 induction by pDCs

To better understand the mechanisms behind the increases in p24 and CD69 expression in T cells cocultured with pDCs, we investigated whether soluble factors secreted by pDCs or their physical contact with CD4 T cells were mediating the increased p24 and CD69 expression. We first blocked ICAM-1 on infected TEM given they showed the highest level of p24 and CD69 induction by pDCs. However, this block did not inhibit p24 (Fig 5A) or CD69 upregulation suggesting that pDC mediate their effects on T cells predominantly via soluble factors and not cellular contact. Therefore, we substituted pDCs with the addition of filtered culture supernatants (depleted of cell-free HIV but retaining the soluble factors) from HIV infected TEM-pDC cocultures or pDCs. Both supernatants upregulated p24 (Fig 5B) and induced CD69 expression (Fig 5C), in similar patterns to cultures where pDCs were added confirming a soluble factor was responsible for their induction. Upon addition of rIFNα8, we observed a significant increase in p24 and CD69 expression (Fig 5D and 5E). Conversely, blocking IFN-I signaling in TEM-pDC cocultures with neutralizing antibodies to IFNα/β and the IFNR significantly decreased p24 and CD69 expression. Addition of rTNFα slightly decreased p24 expression and had no effect on CD69 induction and reciprocally blocking TNFα or its receptor did not significantly affect p24 or CD69 expression. This indicates that IFNα is at least partially responsible for the increased p24 expression and CD69 induction on TEM. Similar patterns of p24 and CD69 induction were detected on TCM and TTM but to a lesser degree than TEM. In addition, when fresh allogeneic pDCs were used in TEM-pDC cocultures (n = 6), we detected a similar magnitude of increased p24 in T cells as in frozen autologous co-cultures, with no statistical difference. However, CD69 induction was induced at a much higher level in allogenic cocultures due to the mixed lymphocyte reaction (MLR) nature of the experiments.

To investigate how pDCs increased p24 expression in resting TEM CD4 T cells, we blocked HIV binding and entry via the CCR5 inhibitor maraviroc for 2 hours before adding pDCs to

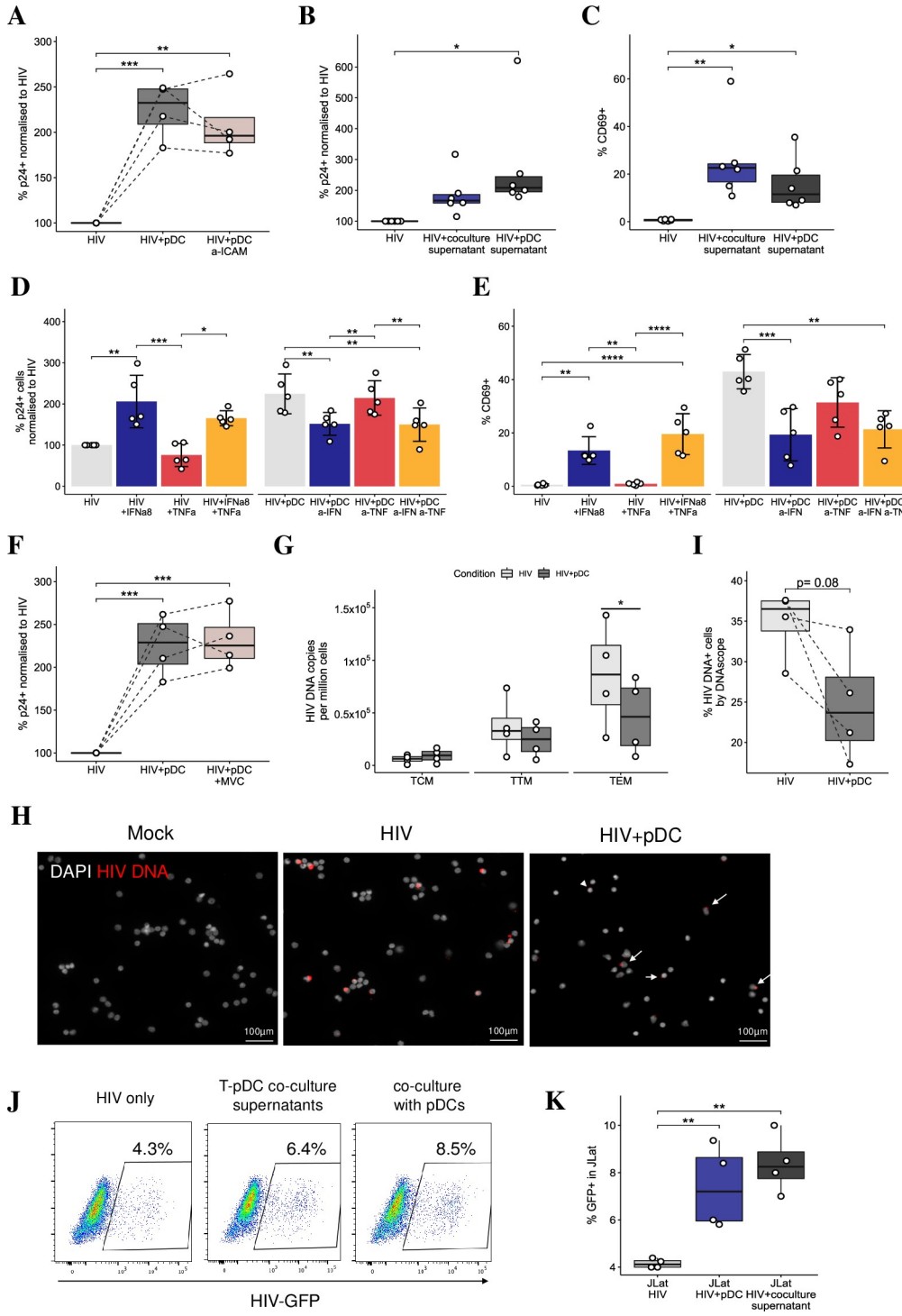

**Fig 5. Mechanisms of p24 and CD69 induction by pDCs.** (A) Normalised p24 expression in resting TEM cells infected with HIV and treated with neutralising antibody to ICAM-1 for 2 hours prior to addition of pDCs (n = 4). (B-C) The effect of virus depleted supernatants (from TEM-pDC cocultures or pDCs) in modulating p24 and CD69 expression in HIV-infected TEM cells (n = 6). (D) Normalised p24 expression and (E) percentage of CD69 expression in resting TEM cells (n = 5) upon modulation of IFN-signaling and/or TNF-signaling via: the addition of exogenous IFNα8 and/or TNFα; blocking the IFNAR1, IFNα and β and/or TNFR1 and TNFα via neutralizing antibodies (a-IFN and a-TNF). (F) Normalised p24 expression in resting TEM cells infected with HIV and treated with maraviroc for 2 hours prior to addition of pDCs (n = 4). (G) Detection of integrated HIV DNA by PCR in infected memory CD4 T cell subsets in the absence (HIV) and presence of pDCs (HIV+pDCs) (n = 4). (H-I) Detection of TEM DNA[+] cells via

DNAScope in the absence (HIV) and presence of pDCs (HIV+pDCs) (n = 4). (J-K) Reversal of viral latency in J-Lat cells measured as increased HIV GFP expression upon co-culture with pDCs or after treatment with supernatants derived from TEM-pDCs. For all data, $^*$p < 0.05, $^{**}$p < 0.01, $^{***}$p < 0.001, $^{****}$p < 0.0001 by repeated measures ANOVA with Tukey post-hoc test (A-F) or paired two-sided t-test (G).

infected TEM. This had no effect on HIV p24 expression (Fig 5F) and suggested that pDCs were not enhancing *de novo* infection and spread but are instead possibly reactivating latent HIV infection in CD4 T cells. To further confirm that HIV spread does not contribute to the increased p24 expression, we assessed integrated HIV DNA at 5 dpi in CD4 T cells (as copies per 1 x $10^6$ cells) using real-time nested Alu-PCR. In the absence of pDCs, there was an increase in integrated HIV DNA from day 2 to day 5 pi which was greater in TTM and TEM compared to TCM. However, upon coculture with pDCs, there was a significant reduction in the number of integrated DNA copies in TEM at 5 dpi compared to HIV alone indicating a reduction in the proportion of cells containing proviral DNA (Fig 5G). Using DNAScope to visualize *in situ* viral DNA in TEM, we observed a trend towards a similar comparative decrease in DNA$^+$ cells upon pDC addition (Fig 5H and 5I), suggesting that pDCs are limiting viral spread in the cell sheet overall, despite the increase in p24 expression. We also performed flow cytometry for the percentage of p24$^+$ protein expressing cells concurrently in 3 of 4 of the individuals with DNAScope results. As a measure of the amount of reactivated virus compared to the total latent pool, we calculated the ratio of p24$^+$ cells by flow cytometry to DNA$^+$ cells by DNAScope. We observed a significantly higher p24$^+$:DNA$^+$ ratio in infected T cells upon coculture with pDCs (S6 Fig), thus further supporting the reactivation of latent HIV by pDCs. Finally, to confirm that HIV reactivation from latency was mediated by pDCs, we used the latently infected J-Lat cells (clone 10.6) and exposed them to supernatants derived from infected TEM-pDC co-cultures or co-cultured them with pDCs. At 2 dpi, we detected a significant increase in HIV GFP expression indicating that pDCs are indeed reactivating latency in J-Lat cells (Fig 5J and 5K).

## pDCs induce a tissue retention phenotype in effector memory CD4 T cells

The expression of the T cell survival marker CD127 (IL-7 receptor), tissue retention markers (CD69 and to a lesser extent PD1 and CD103), combined with the upregulation or downregulation of specific tissue homing markers (CCR7, CXCR6, CXCR3) are identified as the core-signature of CD4 TRM cells in peripheral tissues [14,15] and contribute to their persistence, retention and functionality. Since CD69 and PD-1 were upregulated in all infected resting CD4 T cell subsets cocultured with pDCs, we investigated whether pDCs induce blood CD4 T cells to adopt a pro-retention phenotype. We detected limited CD103 expression on any blood CD4 T cell subsets across all culture conditions. However, we observed a significant upregulation of the residency-associated proteins CD127 and PD-1, and variable expression of CXCR6 on total blood CD4 T cells cocultured with pDCs (Fig 6A). When we looked at retention marker expression on TEM in the pDC cocultures, split by CD69 expression, we found CD127, PD1 and CXCR6 were all significantly upregulated in the CD69$^+$ TEM compared to the CD69$^-$ counterparts (Fig 6B). The tissue homing marker CXCR3 was upregulated on CD69$^+$ cells while CD62L, which is usually downregulated to allow mobilisation to the periphery, was expressed less on CD69$^+$ T cells. In order to see whether this profile adopted by CD69$^+$ TEM following culture with pDCs matches a 'bona-fide' retention phenotype, we investigated the expression of these markers on tissue-derived CD4 T cells, using CD69 as an established marker for CD4 TRM. Given CD4 TRM have been poorly described in anogenital tissue, the primary site of initial HIV infection, we phenotyped CD4 T cells isolated from either

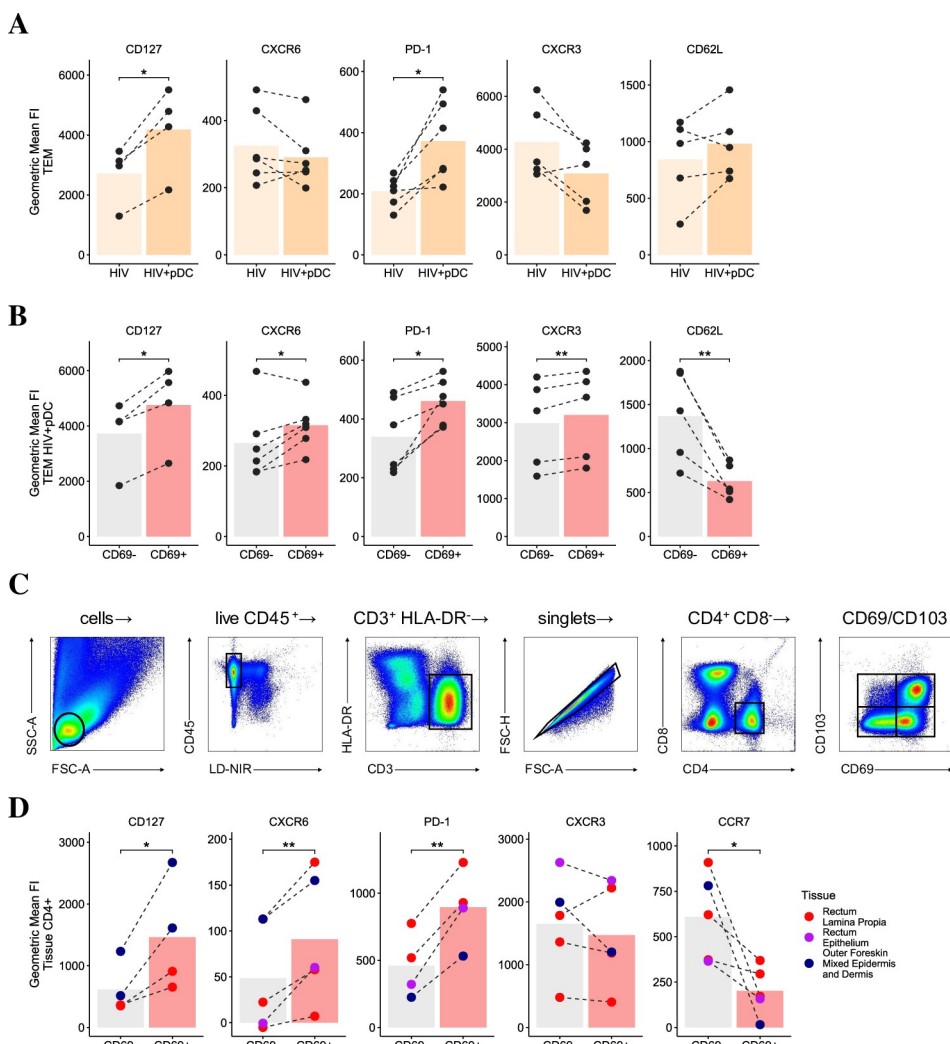

**Fig 6. Induction of tissue retention signature in effector memory CD4 T cells by pDCs.** The geometric mean FI of residency-associated proteins (CD127, CXCR6, PD1) and non-resident markers (CXCR3 and CD62L) in (A) total HIV infected CD4 TEM cocultured with pDCs and (B) in CD69+ versus CD69− expressing CD4 TEM co-cultured with pDCs (n = 4–6 respectively). (C) Representative gating strategy identifying CD69+ tissue resident CD4 T cells isolated via enzymatic digestion of a human rectum. (D) The geometric mean FI of residency-associated proteins (CD127, CXCR6, PD1) and non-resident markers (CCR7, CXCR3) in CD69+ and CD69− expressing CD4 T cells isolated from human rectal epithelium (n = 1), rectal lamina propria (n = 3) and outer foreskin mixed epidermis and dermis (n = 3) tissues. For all data, *p < 0.05, **p < 0.01 by paired two-sided t-test.

rectal tissue or foreskin (Fig 6C). We detected an increased expression of TRM-associated markers CD127, CXCR6 and PD-1 on CD69+ T cells compared to the CD69− cells and a decrease in the peripheral homing marker CCR7 (Fig 6D), while CD62L was not readily detected. Altogether, the phenotype of pDC-induced CD69+ T cells and tissue CD69+ T cells was comparable, thereby suggesting the induction of CD69 on T cells by pDCs may promote their retention in anogenital tissues and limit their egress and dissemination to other tissue sites.

## Discussion

HIV has developed a repertoire of immune evasion strategies, including the ability to shut down IFNβ production in its target cells: DCs, macrophages and T cells [2–4,30]. Although

not present in resting mucosa, pDCs are one of the first cell types to migrate upon inflammation and viral challenge. In macaques infected with SIV intravaginally, pDCs migrate to form a continuous cell layer under the exposed endocervical epithelium [31]. As pDCs produce type I and III IFNs in response to live and inactivated HIV *in vitro* [32,33], they are thought to compensate for the loss of IFNβ production by other HIV target cells and thus help protect the host against the virus. To simulate the role of pDCs in early HIV infection, MDDCs, MDMs and different resting memory CD4 T cell subsets were infected *in vitro*, followed by the addition of pDCs 48h later, as this corresponds to their observed influx into the genital mucosa [10,31].

We have detected expression of all IFN-I and III subtypes by RNAseq upon pDCs exposure to cell free HIV. Key IFN subtypes were also detected by PCR and ELISA in cultures containing pDCs and target cells, indicating that pDCs are the major cellular source of IFNs in cocultures. We have shown that IFNα and TNFα reduced HIV infection of MDDCs while only IFNα was responsible for the reduced infectivity in MDMs (Fig 2E). This intracellular p24 inhibition was via ISG induction, cell maturation and downregulation of CCR5 expression. The mechanism of pDC induced MDDC maturation in our study was via IFNα and TNFα, as previously reported [23,34]. Although HIV manipulation of DC function for viral transfer to T cells is well appreciated [35], much like DCs, macrophages can take up HIV and efficiently transfer virus to CD4 T cells through virological synapses [36]. Macrophages have been found to be significantly enriched in the vaginal mucosa of rhesus macaques in vaginal SIV infection and appear to create concentrated chemokine gradients that recruit pDCs and CD4 T cells to the mucosa [10], thus fuelling the early viral dissemination that leads to systemic infection. Second, the density of CCR5-expressing macrophages increases in the lower rectum which experiences an increased likelihood of HIV transmission [37]. Thus, the reduction of infected DCs and macrophages by pDCs may represent a crucial mechanism through which viral spread to resident or infiltrating memory CD4 T cells in the anogenital mucosa is limited.

HIV infected resting memory CD4 T cell subsets are long-lived latently infected cells which represent the main barrier for clearing HIV infection from patients under ART. When we examined *de novo* HIV infection of the resting memory CD4 T cell subsets, TCMs were observed to have both the lowest integrated HIV DNA copies and the percentage of p24$^+$ cells while TTMs and TEMs contained significantly higher levels of integrated HIV DNA and supported higher levels of p24 expression (Figs 5G and 3C). This correlated with their CCR5 expression and the more advanced differentiation status of TEM where epigenetic changes promote a progressive loss of transcriptional regulation leading to higher levels of gene transcription [38,39]. Valle Cusso et al showed that glucose transporter-1 (Glut1) expression increases from naïve to TCM and TEM and that the susceptibility of CD4 T cell subsets to HIV infection matches their Glut1 expression and metabolic activity as HIV selectively infects highly metabolic CD4+ T cells, independent of their activation phenotype [40]. This is consistent with the differential and increased level of HIV infection we observed in TCM, TTM and TEM in the absence of pDCs. Our data also complimented a study of the CD4 T cell reservoir in patients under long-term ART showing that TEM cells contain the most genetically intact provirus out of the three memory subsets [41]. Altogether, the results of this study and others position TTM and TEM as the major memory subsets housing the HIV reservoir. Furthermore, TEM represent a preferential HIV target during initial mucosal infection given their high propensity for infection, and their abundance in cervicovaginal tissue relative to the other memory CD4 T cell subsets [42].

In contrast to the reduced HIV infectivity in MDDCs and MDMs cocultured with pDCs, surprisingly we detected that pDCs increased intracellular HIV p24 expression (Fig 3B, 3C and 3E), but not extracellular viral production, in resting memory CD4 TCM, TTM and TEM.

This increase occurred in a subset specific manner without any changes in CCR5 expression and activation status except for CD69 induction. Blocking HIV entry did not affect the increased p24$^+$ expression in the CD4 T cells cocultured with pDCs. In cocultures, there was a relative decrease in copies of total integrated HIV DNA by PCR and also in the detection of total DNA$^+$ CD4 T cells by microscopy using DNAScope (Fig 5G and 5I) compared to HIV alone. The decrease in integrated HIV DNA was not due to cell proliferation as we did not detect any cell proliferation using both Cell Trace FarRed and the proliferation marker protein Ki-67. Altogether, this supports a mechanism whereby the increase in p24$^+$ cells stems from the reactivation of latent HIV rather than by enhanced HIV spread. We have demonstrated that increased HIV protein expression is mediated by soluble factors secreted by pDCs and not dependent on cell-to-cell contact as the addition of exogenous rIFNα to HIV infected CD4 T cells also increased p24 expression and, conversely, neutralizing IFN-I signaling significantly decreased the number of p24$^+$ cells. In addition, via type I IFN, pDCs also stimulated CD69 expression on CD4 memory T cells which acts as an inhibitor of CD4 T cell egress [43], and to a lesser extent, modulated the expression of other surface markers characteristic of CD4 TRM cells. This suggests that pDCs enhance the tissue retention of CD4 TEM, thus possibly trapping infected CD4 T cells in peripheral tissue sites exposed to virus [14]. In summary, IFN production by pDCs limited viral spread, reactivated latent provirus and induced a tissue retentive phenotype in CD4 T cells.

One of the key points of interest in this study was understanding the phenotypic changes in CD4 T cells upon coculture with pDCs. CD69 is an early activation marker for T cells and is upregulated within 1 hr of TCR stimulation, with its expression typically peaking 18–24 hours after activation and then declining afterwards. CD25 and CD38 are expressed at 24 hours and their peak expression is detected at 48–72 hours [44], indicating full activation of CD4 T cells. Here, pDCs induced high levels of CD69, very low expression of CD25, CD38 and HLA-DR on both bystander and infected resting TCM, TTM and TEM cells, thus maintaining them in a predominantly resting state (Fig 4D). As CD69 was previously shown to be directly induced by IFNα/β [43,45] it may also be considered as an ISG, in addition to its activation and tissue retention characteristics. When we tested whether CD69$^+$ cells failed to progress to full activation phenotype due to an exhausted status, we detected only low levels of PD-1 in the presence of pDCs but no other exhaustion markers such as TIGIT, Tim-3 and LAG3 indicating that these cells were not progressing into exhaustion. Altogether, this suggests that the enhanced p24 expression in resting memory CD4 T cell subsets do not appear to be related to increased cellular activation and exhaustion was not responsible for the apparent partial activation.

The induction of high CD69 and low PD1 expression on CD4 T cells by pDCs, without expression of other activation markers, is a phenotype reminiscent of CD4 TRM. When we compared TRM markers on CD4 TEM in cocultures with those of CD4 T cells isolated from human anogenital tissues, CD127, CXCR6 and PD1 were significantly upregulated on CD69$^+$ cells in both cocultures and tissue derived CD4 T cells compared to their CD69$^-$ counterparts. However, CXCR3 showed a minor non-significant downregulation on CD69$^+$ cells in anogenital tissues while being upregulated on TEM in cocultures, similar to the upregulation seen on CD69$^+$ CD4 T cells derived from brain [46]. Upregulation of CXCR3 may be involved in mobilising circulating T cells, with T cell infiltration of mucosal tissues shown to be highly dependent on CXCR3 expression in Herpes Simplex Virus-2 infection of the vaginal mucosa [47]. However, once these cells reach the mucosa, its expression may no longer required for retention, and may instead be downregulated as evidenced on the CD69$^+$ cells we isolated from anogenital tissues. CD62L was significantly downregulated on CD69$^+$ TEM cells in cocultures indicating that these cells are also less likely to be recruited back into circulation. CD62L was not detected on tissue derived CD4 T cells as it may be either cleaved by the

enzymes during the process of liberation or not highly expressed in the tissue. CCR7 was downregulated on CD69+ CD4 T cells derived from anogenital tissues and was not detected on TEM in cocultures as TEM were cell sorted as CCR7 negative cells. The lack of CD103 detected on CD4 T cells cocultured with pDCs may be explained by the dependency of CD103 expression on signals from the local tissue environment, with expression only observed once cells have migrated and been sufficiently retained at peripheral sites. For example, the skin is rich in TGFβ and direct interaction of T cells with keratinocytes has been shown to drive CD103 expression in a TGFβ-dependent manner [48]. The upregulated CD127 expression on CD69+ cells in TEM-pDCs and tissue derived CD4 T cells also suggests that pDCs may enhance CD4 T cell survival and function. CD127 has been previously implicated in homeostatic expansion [49] and has been shown to be highly expressed on T cells from ART treated HIV patients with strong immune recovery and CD4 T cell count [50].

IFNs are known to induce restriction factors that interfere with the HIV life cycle. Some factors act before integration to inhibit viral entry (IFITM1-3 ([51]) and transcription (TRIM5a [52], APOBEC3G [53]) thus preventing spread and infection of bystander cells. Other factors act post-integration to limit HIV transcription (TRIM22 [54]) and viral release (IFIT1-3 [5], Viperin [3] and Tetherin [55,56]). Here, we observed that despite an increase in intracellular p24 expression in T cells, the levels of extracellular p24 detected in supernatant were not significantly altered by pDCs (Fig 3F and 3G). This suggests a potential block of HIV at the level of viral assembly or egress. This is consistent with the increased Tetherin expression (S3C and S3D Fig) and sub-membranous colocalization with p24 in memory CD4 T cells induced by IFNα secreted by pDCs (Fig 3H and 3I). This increased expression and colocalization of Tetherin would reduce the release of budded HIV-1 virions from infected cells [55,56] and therefore inhibit further CD4 T cell infection and viral spread. The reduction in extracellular virus infectivity was unlikely to be due to IFNα induced APOBEC3G as our sequencing analysis showed no evidence for APOBEC3G mediated hypermutation in proviral HIV DNA elicited by pDCs. Rather the reduction in infectivity was likely due to the direct antiviral effect of type I IFN on the adjacent cells. Furthermore, despite similar levels of extracellular HIV in the absence and presence of pDCs, the proportion of total HIV DNA+ cells was relatively decreased across all memory T cells when pDCs were present, indicating that pDCs mediate an additional block in *de novo* infection of bystander cells (Fig 5G–5H). Given that HIV infection leads to a chemo-attractive gradient that recruits CD4 T cells into viral sites [10,31], the protection that pDCs provide these newly immigrant cells from viral spread likely plays an important role in restricting local and early HIV infection.

Another concept that we described in this paper was the reactivation of latently infected CD4 T cells by pDCs, with the greatest increase in p24+ cells occurring in the TEM subset. This concept of pDC-driven reactivation has support in the literature, as pDCs and IFN have been investigated as strategies for eradicating the latent reservoir, complementing other latency-reversing agents [21]. Our study investigating early HIV infection and how pDCs via IFN may limit early viral reservoir formation is consistent with a recent study [57] monitoring the pDC response to initial low-level viral replication after ART interruption in HIV infected patients, and thus mimicking the events of early acute infection, which is very challenging to measure. IFN production by pDCs was inversely correlated with viral load and consistent with our data for both myeloid and lymphoid cells *in vitro* and strongly suggesting that early IFN production by pDCs reduces HIV infection, and viral reservoir formation. Furthermore, long-term non-progressors with chronic low-level infection have higher numbers of pDCs compared to healthy noninfected controls [58,59]. It is also complementary to another study by Van der Sluis et al., assessing the effects of IFN on an established latent viral reservoir. They showed diverse effects of IFN on HIV latency: i) inhibition of the establishment of latency via

type I IFNα, IFNβ and IFNω but not IFNε or type III IFNλ1 and λ3; and ii) once latency was established, IFNα but no other IFN subtypes were able to efficiently reverse it in both an *in vitro* model and in CD4 T cells collected from patients on suppressive ART. The mechanism of IFNα mediated HIV reactivation was mediated by phosphorylation of STAT5 [60], but not via the activation of the NF-κB signalling pathway [61]. This is similar to our findings in this study where treatment with exogenous rTNFα, known to activate NF-κB to enhance pro-viral transcription [62] and neutralizing antibody to TNFα had no effect on the proportion of p24 expressing cells. As TCM, TTM and TEM do not express the TNFR [63], this correlates with the lack of rTNFα effect on inducing CD69 or upregulating p24 in our resting CD4 T cell subsets. We have commenced a broad investigation of other mechanisms of HIV reactivation induced by pDCs and type I IFN, using single cell RNA sequencing. We have compared our preliminary data with other transcriptomic studies of HIV reactivation induced by latency reactivating agents (LRA) in CD4 T cells. We detected both overlapping and unique differentially expressed genes, likely coding for pathways leading to NF-κB activation, effects on Trans- Activator of Transcription (Tat) and viral protein U (vpu), enhancement of transcriptional elongation and also coding for long non-coding RNAs which may modulate HIV infection [62]. We are currently confirming the downstream expression of these genes/proteins and their relative importance in initial reactivation compared to those simply facilitating viral replication. Although our transcriptomic studies did not show any Glut RNA signal, this potential mechanism for reactivation will be examined in future.

IFN-I limits viral spread by promoting apoptosis of infected cells and inducing ISGs in surrounding healthy cells to prevent their infection. Our *in vitro* data shows that IFNα-secreting pDCs compensate for the specific inhibitory effects of HIV on IFN-I production in its target cells, macrophages and dendritic cells, by inhibiting HIV infection and protein expression (Figs 1 and 2). Conversely, they induce i) HIV reactivation in CD4 memory T cells without HIV spread (Figs 3 and 5), ii) and a phenotype which probably anchors these cells in tissues throughout the body including anogenital mucosa (Figs 4 and 6) iii) while upregulating MHC-I. Thus, these HIV infected CD4 T cells which have been reactivated from latency by IFNα would be exposed to increased CD8 T cell surveillance. As it is crucial to understand the role of TRM CD4 T cells in HIV pathogenesis and viral reservoir formation, we are currently extending this study by isolating TRM CD4 T cell subsets from a wide range of anogenital tissues to assess which subsets are latently or productively infected and to quantify their HIV infection levels. Confirmation of these findings may help develop novel immunotherapeutic approaches for HIV eradication. For example, using a combination of IFN to reactivate latent cells plus adoptive immunotherapy with CD8 T cells or those expressing chimeric antigen receptor (CAR) could provide a novel 'kick and kill' approach to eradicate HIV latent reservoir in T cells.

## Material and methods

### Ethics statement

This study was approved by the Western Sydney Local Area Health District (WSLHD) Human Research Ethics Committee (HREC); reference number (4192) AU RED HREC/15 WMEAD/11.

### Generation of blood derived MDDCs and MDMs

Peripheral blood mononuclear cells (PBMCs) were isolated via Ficoll-Paque (GE Healthcare Life Sciences, Little Chalfont, United Kingdom) density separation from HIV-seronegative blood supplied by the Australian Red Cross Blood Service, Sydney, Australia. CD14$^+$ cells were

positively selected using a Human CD14 Microbead kit (Miltenyi Biotec, San Diego, CA, USA). To generate MDDCs, CD14$^+$ cells were cultured at 0.5 million cells/mL in RPMI (Lonza, Basel, Switzerland) supplemented with 10% fetal calf serum (FCS; Sigma-Aldrich, St Louis, MO, USA), human interleukin 4 (IL-4; Miltenyi Biotec) and Granulocyte-macrophage colony-stimulating factor (GM-CSF; Miltenyi Biotec) at 50μg/mL each for 6 days. Cytokines were replenished at 25μg/mL on day 3. On day 6, MDDCs were washed twice and resuspended at 3 million cells/mL of RF10 (RPMI+10% FCS) containing cytokines at 50μg/mL prior to infection. To generate MDMs, CD14$^+$ cells were cultured at 1 million cells/mL of RPMI in 24 well plate for 2 hours to promote adherence. Supernatant was then removed and RPMI supplemented with 10% heat inactivated human AB serum (RH10; Sigma-Aldrich, St Louis, MO, USA) was added. Cells were left to differentiate for 5 days prior to HIV infection.

## Isolation and cell sorting of blood CD3$^+$ T cells into activated and resting memory CD4 T cells subsets

PBMCs were depleted of CD8$^+$ T cells using a human CD8$^+$ isolation kit (Miltenyi Biotec). CD3$^+$ T cells were then selected via the human CD3 Microbead kit (Miltenyi Biotec). To isolate the different subsets of memory CD4 T cells (central memory (TCM), transitional memory (TTM) and effector memory (TEM)), the CD3$^+$ CD8$^-$ T cells were stained with Fixable Near-Infra Red (LDNIR, Invitrogen, Massachusetts, USA) for 30 minutes at 4°C followed by a wash with MACS buffer [PBS, 1% (v/v) human AB serum and 2 mM ethylenediaminetetraacetic acid (EDTA, Sigma-Aldrich)]. Cells were then resuspended in Brilliant Stain Buffer (Becton Dickinson (BD), New Jersey, USA) and stained for 30 minutes at 4°C with the following antibodies: CD3 (AF700, clone UCHT1), CD4 (BUV496, clone SK3), HLA-DR (BB515, clone G46-6), CD25 (BV711, clone MA251), CD38 (BV421, clone HIT2), CD69 (PECy7, clone FN50), CD45RO (BV605, clone UCHL1), CCR7 (BB700, clone 3D12) and CD27 (BUV395, clone L128). All antibodies were purchased from BD Biosciences or BioLegend. This was followed by a wash in MACS buffer and resuspension at 10$^8$ cells/mL for cell sorting on the BD Influx Cell Sorter. Sort purity was assessed post cell sorting and was consistently >95%. Resting memory cells were sorted as CD3$^+$CD4$^+$CD69$^-$CD25$^-$CD38$^-$HLA-DR$^-$CD45RO$^+$ (S7A Fig) and were further subdivided into TCM (CCR7$^+$CD27$^+$), TTM (CCR7$^-$CD27$^+$) and TEM (CCR7$^-$CD27$^-$).

## Isolation of pDCs from blood

pDCs were isolated from the CD3 or CD14 negative PBMC population using the Human pDC Isolation Kit II (Miltenyi Biotec) and were cryopreserved using CryoStor CS10 (StemCell) at -80°C for 48 hours prior to their addition to MDDCs, MDMs and CD4 T cells. To exclude contaminating Axl$^+$Siglec6$^+$ myeloid DC (ASDC) populations [64,65], recently reported to be misidentified as pDCs, defrosted pDCs were stained with the following antibodies purchased from BD Biosciences or BioLegend unless otherwise stated: Lin1 (CD3, CD14, CD16, CD20 and CD56; BV510, clone B56), HLA-DR (BV605, clone G46-6), Siglec6 (APC, clone REA852, Miltenyi Biotec), Axl (PECy7, clone DS7HAXL, ThermoFisher), CD123 (BV421, clone 7G3) and BDCA2 (Vio-bright FITC, clone REA693, Miltenyi Biotec). Cell sorting was performed to exclude Axl$^+$ Siglec6$^+$ expressing cells, with 'bona-fide' pDCs used in our cocultures being Lin1$^-$HLA-DR$^+$Axl$^-$Siglec6$^-$BDCA2$^+$CD123$^+$ cells with purity >95% (S7B Fig).

## HIV Infection of MDDCs, MDMs, pDCs and CD4 T Cells

Purified high-titre stocks of the CCR5 tropic laboratory adapted HIV$_{BaL}$ with a 50% tissue culture-infective dose of 5×10$^8$/mL were produced with the use of tangential filter concentration

to eliminate cellular HIV DNA, proteins and cytokines as described previously [66]. The clinical transmitted founder isolate $HIV_{Z3678M}$ ($HIV_{TF}$, provided by Eric Hunter, Genbank accession number: KR820393) was generated as previously described [11]. The endotoxin levels of these virus stocks were less than the detectable limit of 0.005 U/ml or 0.0005 ng/ml (Limulus amebocyte lysate assay; Sigma-Aldrich). MDDCs, MDMs and pDCs were infected with a MOI of 2 due to their expression of HIV restriction factors while CD4 T cells were exposed to a MOI of 0.75 given the cytopathic effects of HIV on the latter and based on the viral titration of different MOIs (0.5, 0.75 and 1) in PHA/IL-2 activated CD4 T cells and the resting TCM cell subset. At 72 hours pi, when we compared MOIs of 0.5 and 0.75, we observed increased p24 expression in the activated CD4+ T cells and TCM cells. However, p24 expression plateaued with the MOI of 0.75 and 1 in both the activated CD4 T cells and the resting TCM cells. When MOIs below 0.75 were used, we did not detect any p24 infection in TCM derived from 75% of our donors. As HIV infection and DNA integration is much lower in TCM compared to TTM and TEM, the MOI of 0.75 was used for the subsequent infection of T cells.

Mock-treated and HIV infected cells were incubated at 37˚C overnight, washed twice with RPMI and then cultured in their appropriate media at 1 million cells/mL of RF10 in 96 U well bottom shaped plates for CD4 T cells and MDDCs while MDMs were kept in 24 well plates.

## Cocultures of pDCs with HIV-infected Cells

Autologous pDCs supplemented with 1ng/mL of IL-3 (Miltenyi Biotec) to promote their viability were added to HIV-infected MDDCs, MDMs or CD4 T cells at a ratio of 1 pDC:3 infected cells. Where indicated, MDDCs, MDMs and CD4 resting memory T cell subsets were treated with reagents including cytokines, neutralising antibodies or HIV drugs either individually or in combination for 2 h prior to the addition of pDCs. These reagents included: an inhibitor of the intercellular adhesion molecule and component of the immunologic synapse (anti-ICAM-1, 10μg/mL, R&D systems, clone BBIG-I1); the CCR5 inhibitor maraviroc (10μM, AIDS Reagent Program); the reverse transcription inhibitor Zidovudine (50μM, AIDS Reagent Program); neutralising antibodies to receptors of IFN (IFNAR, 20μg/mL, PBL Assay Science, clone MMHAR-2) and TNF (TNFR1, 10μg/mL, R&D systems, clone 16803) in addition to neutralising antibodies for secreted IFNα8 (20000U/mL, PBL Assay Science), IFNβ (5000U/mL, PBL Assay Science) and TNFα (1μg/mL, R&D systems) to block signalling pathways of INF and TNF; recombinant (r) human TNFα (10ng/mL, R&D systems) and rIFNα8 (PBL Assay Science) was used at 350ng/mL for MDDCs and MDMs and at a graded concentration of 300, 600 and 900ng/mL for TCM, TTM and TEM based on the amount of IFNα detected by ELISA in infected MDDCs (S1D Fig) and T cell subsets (S5A Fig) upon their coculture with pDCs.

## Assessment of CCR5 expression, myeloid cell maturation, T cell activation and exhaustion, HIV infection, and tissue retention phenotype

MDDCs, MDMs and T cells were collected 5 dpi and stained with LDNIR for 30 minutes at 4˚C followed by a wash with FACS buffer [PBS, 2% (v/v) human FCS, 2 mM EDTA and 0.1% (w/v) sodium azide (Sigma-Aldrich)]. They were subsequently stained for 30 min at 4˚C with antibodies (purchased from BD Biosciences or BioLegend unless otherwise stated) for cell surface markers to assess CCR5: (PECF594, clone 2D7); MDDC and macrophage maturation: CD80 (PECy7, clone L307.4) and CD86 (BV605, clone FUN-1); T cell activation (CD25, CD38, CD69 and HLA-DR; listed above in the section of cell sorting of blood CD3[+] T cells); Exhaustion: CTLA-4 (BV766, clone BNI3), Lag-3 (APC, clone ABCE0515011, R&D Systems), PD-1 (BV480, clone EH12.1) and TIM-3 (BV650, clone F38-2E2); MHC-I (HLA-ABC;

BUV805, clone G46-2.6); Tissue resident markers: CD69 (PECy7, clone FN50l), CD127 (PE Dazzle or BV650, clone AO18D5), CXCR6 (PE Dazzle or APC, clone KO41E5), PD1 (BUV737, clone EH12.1), CXCR3 (AF647, clone GO25H7), CD62L (APC-Cy7, clone DREG-56). They were then incubated with BD Cytofix/Cytoperm for 15 minutes at 4˚C, washed in perm buffer [PBS with 1% (v/v) FCS, 0.1% (w/v) saponin, 0.1% (w/v) sodium azide] followed by intracellular staining with antibodies to the HIV capsid; p24 APC (28b7, Medimabs) and/or p24 PE (KC57, Beckman Coulter) for 30 minutes at 4˚C. Cells were again washed with perm buffer and acquired on the BD LSR Fortessa X-20 (BD Biosciences). Data was recorded using BD FACSDiva (BD Biosciences) and analysed using FlowJo (FlowJo LLC, Oregon, US).

## Library preparation for RNA sequencing and analysis

RNA was extracted from mock and HIV infected pDCs using the High Pure RNA Isolation Kit (Roche, Basel, Switzerland). It was quantified and quality assessed using the Quant-iTTM RNA kit (Invitrogen) and the 2100 Bioanalyser (Agilent Technologies, California, US), respectively. Samples were prepared for RNAseq using the Stranded mRNA Sample Preparation Kit (Illumina, CA, USA), according to the manufacturer's instructions, as per the Low Sample (LS) protocol guidelines. Prepared libraries were sequenced using Illumina HiSeq 2500 (Illumina), generating 50 bp single-end sequences. Raw sequence data was aligned to the University of California Santa Cruz (UCSC) human reference genome (hg38) using Rsubread [67]. Aligned sequencing reads were summarised to counts per gene. Gene counts were normalised across samples using TMM normalisation [68].

## Assessment of IFN production by ELISA

IFNα and β concentrations were assessed in culture supernatants collected 5 days pi using the VeriKine Human IFNα multiple subtype ELISA kit and the VeriKine Human IFNβ ELISA kits (PBL Assay Science, New Jersey, US) respectively, according to the manufacturer's instructions. Colorimetric intensity was determined at 450 nm using the Victor X3 2030 Multi Label Reader (Perkin Elmer). A four-parameter logistic regression standard curve was constructed and used to deduce the concentrations of IFNα and β.

## Assessment of virus production extracellularly

RT activity and p24 concentration were quantified in culture supernatants using the Lenti RT Activity kit (Cavidi, Uppsala, Sweden) and the HIV-1 Gag p24 Quantikine ELISA (R&D Systems, Minnesota, US) according to the manufacturer's protocol.

## Investigating CD69 and p24 induction

To assess whether soluble factors secreted by pDCs were inducing p24 and CD69 expression in CD4 T cells, we separated HIV virions from soluble factors in the supernatants. 50 μl aliquots of supernatants derived from either 8 infected donor CD4 TEM cells in the presence of pDCs (co-culture supernatant) or 8 infected donor pDCs (pDC supernatant) were loaded to 100 kDa molecular weight cut-off filters (Vivaspin 500, Sartorius, Göttingen, Germany). Filters were spun for 15 minutes at 3000 g to retain HIV virions in the filters and collect the flow through fractions containing soluble factors. Day 2 infected TEM or J-Lat clone 10.6 (NIH AIDS Reagent Program) were depleted of their supernatants and replaced with either 160 μl of the co-culture supernatant or 40 μl of the pDC supernatant.

## Quantification of integrated HIV DNA by PCR

CD4 T cells were separated from pDCs in cocultures using the human CD3 Positive Selection kit (Stem Cell Technologies) according to the manufacturer's instructions. CD4 T cells were lysed in 20μl of DNA lysis buffer (10mM Tris-HCL, 0.5% (v/v) NP-40, 0.5% (v/v) Tween 20) containing 1mg/mL of Proteinase K (Sigma-Aldrich, St Louis, MO, USA). Cell lysates were incubated at 56°C for 1 hour, and then for 10 min at 95°C to inactivate Proteinase K. To assess copies of integrated HIV DNA in CD4 T cells, ACH2 cells (AIDS Reagent Program) were used to generate standards with 10-fold serial dilutions ranging from $3 \times 10^5$ to 3 copies of integrated HIV DNA. To quantify the exact number of cells in each sample, standards for CD3 were derived from a CD3/2LTR plasmid (gift from the Chomont Laboratory, Montreal University, Canada) after 10-fold serial dilution to cover the range of $6 \times 10^5$ to 6 copies. HIV integration was assessed by a nested PCR as previously described [69].

## DNAScope and immunofluorescent microscopy

Mock and HIV infected TEM in the presence and absence of pDCs were washed in PBS, fixed, permeabilised in BD Cytofix/Cytoperm for 15 min at RT and washed in perm buffer at 1500 rpm for 5 min. They were then spun onto slides coated with Poly-L-Lysine using the Statspin Cytofuge 2 (Beckman-Coulter, Brea, CA, USA) at 1600rpm for 4 min. They were then used for either the detection of colocalised Tetherin and p24 or HIV DNAScope using the RNAScope 2.5HD Red Reagent Kit (ACD Bio) with custom probes against HIV-1$_{BaL}$ DNA (ACD Bio) as per the RNAScope 2.5HD Manual protocol. For DNAScope, slides were subject to protease treatment for 20 min at 40°C in a HybEZ hybridisation oven (220 VAC), followed by a 2 h incubation with the HIV$_{BaL}$ DNA probes at 40°C. Slides were then treated with 6 rounds of amplification and incubated for 3 min with detection reagents provided with the RNAScope kit. Cells were then stained with rabbit CD3 (clone A052, DAKO) conjugated to Cyanine7 (Lumiprobe) to identify TEM cells from pDCs. All images were acquired on the Olympus VS120 Slidescanner at 20x objective using Olympus VS-ASW (version 2.9) software. CD3$^+$ DNA$^+$ cells from 4 donors were then manually counted in ImageJ from five random fields of view. For co-localisation of Tetherin and p24 within cells, a blocking solution of PBS supplemented with 1% BSA (w/v), 0.1% saponin (w/v) and 10% donkey serum (v/v) (Sigma-Aldrich) was added to slides for 30 minutes at room temperature. A primary antibody cocktail made up in the blocking solution containing mouse anti-p24 (KC57) conjugated to FITC, mouse anti-Tetherin (RS38E) conjugated to AF647 and rabbit anti-CD3 (polyclonal) conjugated in house to Cyanine-3 (Luminoprobe) was added to the slides for 1 hour at room temperature. Slides were then DAPI stained and mounted using SlowFade Diamond Antifade (Invitrogen). Cells were imaged using x100 objective on the Deltavision deconvolution microscope with a Photometrics CoolSnap QE camera (GE Healthcare, IL). Multiple z-planes were acquired using 0.1um optical section spacing with a total thickness between 7-9um (70–90 sections) per image. Analysis was performed using Fiji (2.0.0 version).

## Tissue processing

Healthy human foreskin and rectum were obtained within 30 min of surgery from consented adult patients (>18 years of age) undergoing circumcision (foreskin) or colorectal surgery (rectum) with written consent from all donors. Fat and submucosa were removed from the tissue and CD4 T cells were isolated using optimised collagenase digestion protocols as previously described [11]. Briefly, foreskin and rectal tissues were cut into smaller pieces. Foreskins were placed in RPMI containing 100U/ml DNase I (Worthington Industries, Columbus, OH, USA) and 200 U/ml collagenase Type IV (Worthington Industries) and incubated for 120

minutes at 37˚C to collect epidermal and dermal cells. Rectal mucosa was placed in RF10 containing 0.15% dithiothreitol (Sigma-Aldrich) and 2mM Ethylenediaminetetraacetic Acid (EDTA; Sigma-Aldrich) followed by an incubation for 15 minutes at 37˚C to strip epithelial contents. Epithelial cells were collected, and this stripping process was repeated once more. The remaining tissue was placed in DNase I /collagenase Type IV solution (as above) and incubated for 30 minutes at 37˚C to collect lamina propria cells. The collagenase digestion was repeated once more. The cell pellets collected from two rounds of collagenase treatment were resuspended in RPMI. Cells collected from either foreskin (mixed epidermis and dermis), rectum epithelium or rectum lamina propria were spun on a Ficoll-Paque gradient (GE Healthcare Life Sciences) and collected from the Ficoll-PBS interface. They were then stained with Fixable Viability Stain 700 (FVS700, Thermo Fisher Scientific) and cell surface antibodies for: CD45 (BV480, clone HI30), HLA-DR (BUV395, clone G46-6), CD3 (BUV496, clone UCHT1), CD4 (BV785, clone OKT4), CD8 (PerCP Cy5.5, clone SK1), CD69 (PE Cy7, clone FN50l), CD103 (BV711, clone Ber-ACT8), CCR7 (BV421, clone 150503), CD127, CXCR6, PD1, CXCR3, CD62L (all listed above) to phenotype them for tissue retention markers by flow cytometry.

## Single-proviral sequencing

HIV-DNA sequences were obtained at limiting dilution by single-proviral sequencing using primers flanking the env region (V1–V3; 813 base pairs) as previously described [70–77]. A total of 72 and 130 individual HIV-DNA *env* V1-V3 sequences were obtained from Donor 1 and Donor 2, respectively. These sequences were used to construct maximum likelihood phylogenetic trees with generalized time-reversible nucleotide substitution model [78,79]. The sequences containing premature stop codons were identified using Geneious version 8.1.9 [80]. HIV-DNA sequences containing G to A and non-G to A mutations were identified by comparing to HIV-1 BaL (accession: AY426110.1). All HIV-DNA sequences obtained by SPS were screened for G to A hypermutation by using the Los Alamos Hypermut tool [81]. The frequency of G to A mutations within individual HIV-DNA sequences was compared to the frequency of non-G to A mutations using a paired T-test (Stata 15, StataCorp. 2017. Stata Statistical Software: Release 15. College Station, TX: StataCorp LLC).

## Statistics

Statistical comparisons between two groups were performed using Wilcoxon signed-rank tests unless otherwise stated. In cases where paired two-sided t-tests were used, assumptions of normality were first tested using a Shapiro-Wilk test. For correlation between CD69 and p24, a linear mixed model using the lme4 and lmer Test functions in R, treating subset as a fixed variable. One-sided Tests were used where the direction of the result was expected with IFN induction of Tetherin expression. For comparisons between more than two groups, a repeated measure one-way analysis of variance (ANOVA) with Tukey's post-hoc test was conducted. The frequency of G to A mutations was conducted via unpaired two sample T test. All statistical analysis was performed in R or Graphpad Prism 8, and $p < 0.05$ was considered statistically significant for all tests; *$p < 0.05$, **$p < 0.01$, ***$p < 0.001$, ****$p < 0.0001$. Error bars represent standard deviation across all analyses.

## Supporting information

**S1 Fig. (**S1A) pDCs induce IFN and ISG expression in MDDC-pDC cocultures. Gene bank accession numbers can be reviewed in Nasr et al (3). n = 4, *$p < 0.05$, **$p < 0.01$ by paired two-sided t-test with multiple testing correction. (S1B) HIV exposed pDCs expressed IFNs but

no significant ISGs mRNA. (S1C) RNAseq data showing TNF subtype gene induction in pDCs at 18 hours post HIV exposure. (S1D) Detection of IFNα and IFNβ by ELISA in supernatants derived from MDDC-pDC cocultures.
(TIF)

**S2 Fig.  (S2A-B) Induction of maturation marker on MDMs cocultured with pDCs.** The median FI of maturation markers CD80 (B-upper panel) and CD86 (B-lower panel) in MDDCs that were either mock, HIV infected MDDCs in the presence and absence of pDCs, HIV infected MDDCs treated with exogenous rIFNα8 and/or rTNFα, HIV infected MDDCs treated with antibodies to blocking IFN and/or TNF signaling in pDC cocultures. Data is shown as normalized to 100% in mock infected cells. n = 5 individuals, $^*p < 0.05$, $^{**}p < 0.01$, $^{***}p < 0.001$ by repeated measures ANOVA with Tukey post-hoc test.
(TIF)

**S3 Fig.  (S3A)** Median fluorescent intensity (FI) of p24$^+$ TCM, TTM and TEM (n = 10) in the absence (HIV) or presence of pDCs (HIV+pDC). (S3B) Assessment of infectious virus release via the TZMbl assay. Graphs show the number of infected cells after exposure to infectious supernatants derived from infected resting TCM, TTM and TEM cells in the absence (HIV) or presence of pDCs (HIV+pDC). n = 10–11 individuals, $^{**}p < 0.01$, $^{***}p < 0.001$ by Wilcoxon signed rank test. (S3C-D) Quantification of Tetherin expression by Imagestream flow cytometry in TEM that were either mock or infected with HIV in the presence and absence of pDCs. n = 3 $^*p < 0.05$ by one tail paired T test.
(TIF)

**S4 Fig. HIV-DNA *env* V1-V3 sequences within CD4+ memory T-cells cocultured with pDC or pDC supernatant.** (S4A) Phylogenetic trees of HIV-DNA sequences within CD4 memory T-cells derived from Donor 1 and Donor 2. For the phylogenetic trees, individual HIV-DNA sequences derived from the memory T-cells infected with HIV-1 BaL (HIV) and the sequences from the infected cells cocultured with pDC (pDC) or supernatant from pDC (Supernatant from pDC) are shown as in the legend. For Donors 1 and 2, HIV-DNA sequences derived from TTM cells were included in the tree (circle). For Donor 2, the viral sequences isolated from TEM cells are shown (square). The outer layer of each phylogenetic tree shows the sequences containing premature stop codons (green square). The sequences without the stop codons are also indicated in this outer layer (pink square). HIV-1 BaL (accession: AY426110.1) was used to root the trees (red triangle). (S4B to D) Frequencies of G to A mutations (blue data points) and non-G to A mutations (pink data points) within individual HIV-DNA sequences obtained from the infected TTM cells (Donors 1 and 2) and TEM cells (Donor 2) exposed to the three experimental conditions.
(TIF)

**S5 Fig.  (S5A)** Detection of IFNα and IFNβ by ELISA in supernatants derived from T cells -pDC cocultures. (S5B-C) pDCs induce IFN and ISG expression in TCM (B) and TTM (C) cocultured with pDC. n = 3 $^*p < 0.05$, $^{**}p < 0.01$ by paired two-sided t-test with multiple testing correction. (S5D-E) Expression of T cell activation markers. Graphs show the percentage of resting T cells expressing CD25 (S5D), CD38 (S5E), HLADR (S5F) and PD1 (S5G) in the absence (HIV) or presence of pDCs (HIV+pDC). n = 11 individuals, $^*p < 0.05$, $^{**}p < 0.01$, $^{***}p < 0.001$ by one-way repeated ANOVA.
(TIF)

**S6 Fig. Ratio between percentage of p24$^+$ cells detected by flow cytometry and HIV DNA$^+$ cells identified by DNAscope in TEM cells infected with HIV in the absence or presence of**

pDCs (n = 4). $^*$p $<$ 0.05 by paired two-sided t-test.
(TIF)

S7 Fig. (S7A) Cell sorting strategy to isolate resting memory CD4 TCM, TTM and TEM cells from blood. After excluding debris, doublets and dead cells, live CD3 CD4 T cells were gated. Resting cells were identified as the negative population of cells that did not express the T cell activation markers HLADR, CD25, CD38 or CD69. Memory cells were gated as CD45RO$^+$ with CCR7 and CD27 then used to discriminate between the TCM (CCR7$^+$ CD27$^+$), TTM (CCR7$^-$ CD27$^+$) and the TEM (CCR7$^-$ CD27$^-$) subsets. (S7B) Cell Sorting Strategy to isolate 'bona-fide' pDCs from blood. By excluding Axl$^+$ Siglec6$^+$ cells, 'bona-fide' pDCs in our co-cultures were Lineage$^-$HLA-DR$^{-/lo}$ Axl$^-$ Siglec6$^-$ BDCA2$^+$ CD123$^+$ cells with purity $>$95%.
(TIF)

## Acknowledgments

Flow cytometry and cell sorting was performed in the Flow Cytometry Core Facility that is supported by The Westmead Institute for Medical Research, Westmead Research Hub, Cancer Institute New South Wales and National Health and Medical Research Council (NHMRC), Australia. ImageStream flow cytometry was performed at The Centenary Institute, University of Sydney, Sydney, New South Wales. We thank Naomi R. Truong and Kerrie J. Sandgren for their assistance in acquiring the Deltavision microscopy image and for providing intellectual input into the analysis.

## Author Contributions

**Conceptualization:** Orion Tong, Anthony L. Cunningham, Najla Nasr.

**Data curation:** Orion Tong, Gabriel Duette, Thomas R. O'Neil, Caroline M. Royle, Hafsa Rana, Blake Johnson, Nicole Popovic, Suat Dervish, Michelle A. E. Brouwer, Sarah Palmer, Eunok Lee, Najla Nasr.

**Formal analysis:** Orion Tong, Gabriel Duette, Thomas R. O'Neil, Caroline M. Royle, Hafsa Rana, Blake Johnson, Nicole Popovic, Michelle A. E. Brouwer, Heeva Baharlou, Ellis Patrick, Sarah Palmer, Eunok Lee, Najla Nasr.

**Funding acquisition:** Anthony L. Cunningham, Najla Nasr.

**Investigation:** Orion Tong, Anthony L. Cunningham, Najla Nasr.

**Methodology:** Orion Tong, Thomas R. O'Neil, Hafsa Rana, Heeva Baharlou, Sarah Palmer, Eunok Lee, Eric Hunter, Andrew N. Harman, Najla Nasr.

**Project administration:** Anthony L. Cunningham, Najla Nasr.

**Resources:** Orion Tong, Grahame Ctercteko, Eric Hunter, Andrew N. Harman, Anthony L. Cunningham, Najla Nasr.

**Software:** Orion Tong, Najla Nasr.

**Supervision:** Orion Tong, Anthony L. Cunningham, Najla Nasr.

**Validation:** Orion Tong, Anthony L. Cunningham, Najla Nasr.

**Visualization:** Orion Tong, Anthony L. Cunningham, Najla Nasr.

**Writing – original draft:** Orion Tong, Eunok Lee, Anthony L. Cunningham, Najla Nasr.

**Writing – review & editing:** Orion Tong, Sarah Palmer, Anthony L. Cunningham, Najla Nasr.

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
