## [Decision Letter · Decision Letter 0]

19 Nov 2020

Dear Dr Nasr,

Thank you very much for submitting your manuscript "Plasmacytoid dendritic cells have divergent effects on HIV infection of initial target cells and induce a pro-retention phenotype" for consideration at PLOS Pathogens. As with all papers reviewed by the journal, your manuscript was reviewed by members of the editorial board and by several independent reviewers. In light of the reviews (below this email), we would like to invite the resubmission of a significantly-revised version that takes into account the reviewers' comments.

We cannot make any decision about publication until we have seen the revised manuscript and your response to the reviewers' comments. Your revised manuscript is also likely to be sent to reviewers for further evaluation.

Sincerely,

Daniel C. Douek

Associate Editor

PLOS Pathogens

Richard Koup

Section Editor

PLOS Pathogens

Kasturi Haldar

Editor-in-Chief

PLOS Pathogens

orcid.org/0000-0001-5065-158X

Michael Malim

Editor-in-Chief

PLOS Pathogens

orcid.org/0000-0002-7699-2064

Reviewer's Responses to Questions

**Part I - Summary**

Reviewer #1: Tong and colleagues performed several experiments to investigate the role of plasmacytoid dendritic cells (pDCs) and their interferon (IFN) production on HIV infection in myeloid and resting CD4+ T cells using coculture models.

They first showed that pDCs coculture reduced % and MFI of p24+ MDMs and MDDCs. This reduction is likely due to the production of IFNa that matures these cells and decreases their CCR5 expression.

Next, they moved to resting CD4+ T cells. Infecting subpopulations of resting CD4+ T cells with HIV-Bal or HIV-TF increased the percentage of cells expressing p24. This induction was associated with an increase of the CD69 early activation marker. However, this induction in %p24+ cells was also associated with no increase in virion production, decreased viral infectivity, and a decrease in HIV DNA levels.

Reviewer #2: Interferons play an important role in regulating viral infections, including in HIV infection. This has been documented in many studies and the impact of Interferon type I is variable depending on many factors. Here the authors propose to study the question from a distinct angle, by analysing the impact of pDC on HIV replication in distinct target cells in vitro. Indeed, instead of adding INF-I, pDC were co-cultured with target cells. A variety of distinct target cells were studied, ie resting central, transitional and effector memory CD4 + T cells as well as myeloid DC and macrophages (all primary cells and no cell lines). Cocultures with autologous pDC resulted in decreased HIV replication in myeloid cells (MDM, MDDC), suggesting PDC infiltrating mucosa rapidly upon infection could limit spread of HIV. In contrast, cocultures with resting memory CD4+ T cells resulted in increased intracellular p24 levels in resting memory CD4 cells, without increases in released extracellular virus. The analyses indicate a reactivation of the virus in the CD4 T cells. This underlies an already suggested role of IFN-a as latency-reversal agent. They showed that soluble factors, including IFN-a, mediate these effects. Furthermore, co-culture with PDC induced a tissue-resident like phenotype of the CD4+ T cells, concomitant with upregulation of MHC-I suggesting that IFN-I promotes retaining of CD4 T cells in tissues and enhances their susceptibility for immune clearance by (functional) CD8+ T cells. The authors are well known for having provided fundamental important insights into the biology of dendritic cells. The manuscript is very clear and well written. The discussion is a little long. This study unifies several concepts discussed in the literature in an elegant way. It would benefit from a few complementary experiments.

**Part II – Major Issues: Key Experiments Required for Acceptance**

Reviewer #1: The manuscript is well-written, and IFN is an important molecule during early and chronic HIV infection. Furthermore, it was recently highlighted in viral rebound after stopping therapy. Therefore, a better understanding of the interplay between pDCs, IFN production, and HIV is warranted.

However, there are some comments and additional experiments that can add more depth to the current version of this manuscript:

1) One of the main messages of the resting CD4+ T cell competent of the study is that pDC (through their production of IFN) reactivate latent infection (evident by an increase in % of p24+ cells even after blocking viral entry). While the results presented in the study may support this hypothesis (especially in light of recent literature), a couple of potential experiments could strengthen this point, including coculture pDCs with latently infected cells (from latency models such as JLat or ACH2 or CD4+ T cells from people living with HIV on suppressive ART) to directly measure latency reactivation mediated by pDCs. Another possibility would be sort p24-negative cells and only coculture these with pDCs and observe latency reactivation. Finally, the MFI of all of these experiments should be presented in addition to % of p24; the MFI should probably also increase per cell basis if pDCs have a reactivation ability.

2) Another important observation was the reduction in viral infectivity (in supplemental figures), matching with an APOBEC activity in mutating progeny virus and rendering them defective. This observation could be added to the main Figures, and the authors could sequence this virus to show if it has an APOBEC G-A mutation signatures. That can add depth to this important point that can highlight the opposing effects of IFN and pDC (on the one hand, reactivate the virus but render some of them defective). Also, the increase in viral expression that does not translate to an increase in viral production is consistent with BST2(therein) activity. Again, quick staining of therein on the same cells, if available, could add more depth to this observation.

3) While the manuscript focused on highlighting early infection events, an important potential implication of it could be during therapy interruption. Recently, it has been shown (Julie L. Mitchell et al., JCI (Plasmacytoid dendritic cells sense HIV replication before detectable viremia following treatment interruption)) that pDC production of IFN inversely correlates with time to viral rebound after treatment interruption. I think that deserves more discussion and how the current data align (or not) with this observation.

4) All infections were performed using HIV-Bal (CCR5 tropic) or a transmitted-founder virus that is also CCR5 tropic. It would have been good to repeat some key experiments with a CXCR4-tropic virus (such as NL43) and show whether the results (for both myeloid or CD4+ T cells) are tropic-dependent or not.

Reviewer #2: The pDC used for the cocultures were previously frozen. While this was done to allow culture with autologous cells, it is known that PDC decrease in function and viability during cryopreservation. It is not clear if fresh PDC would not have an even stronger or distinct effect. The viability and capacity of IFN-I production of fresh and frozen PDC as used in the protocol could be compared or other control experiments could be performed with fresh PDC, such as testing the impact of supernatants from fresh PDC on infected cells. And could it be that some of the effects are mediated by factors released from dying PDC?

The study is also interesting because it tries to reproduce the physiological conditions by using the IFN-I-producing cells instead of adding exogeneous IFN-I to the cultures. Still, very high MOI were used to infect the cells (probably in order to reach a high frequency of infected cells and to be able to detect intra-cellular p24). Such high MOI might in most cases not correspond to physiological conditions and might hide a potential inhibitory effect of the PDC in the CD4 T cells. It would be good to confirm some data with lower MOI. The reduction of viral replication might be measured in that case by more sensitive methods (RT-PCR or p24 Simoa).

**Part III – Minor Issues: Editorial and Data Presentation Modifications**

Reviewer #1: 5) A minor point, it is not clear how the statistical analysis in Figure 4C was done; if it was a simple correlation, that would have been wrong as repeated measures of sane sample donors were used in the plot (different subpopulations of same donors), and that should be adjusted in the statistical analysis. The correlation will obviously stay highly significant, but the p-value might change for scientific accuracy.

6) Another minor point, having the methods following the results, the S1A Fig. should be moved to be the last supplementary Figure as it first appears in the methods after all other supplementary Figures.

Reviewer #2: The lack of detectable productive infection of pDC might be related to low viral production or tto the fact that pDC were cultured apparently for 3-5 days, which might not be sufficiently long for such cells to identify productive infection, in particular if measured by intra-cellular p24 that is not a very sensitive method. Also, pDC do not survive a long time in culture. In addition, it seems that the PDC were frozen, which likely has resulted in reduced viability of the cells. The statement that there was no productive infection in pDC should be moderated and complemented by these limitations for clarification.

Lines 212-220: It was analysed if the increased p24 expression in memory CD4+ T cells cocultured with pDC was not due to a defect in IFN production and/or ISG expression. The results are presented as “data not shown”. The results could be shown as supplementary data as this seems to be an important point.

Can it be excluded that the decrease of integrated HIV DNA is not the result of proliferation (and in consequence of a dilution of infected cells within total cells) and/or of more de novo infections (which are generally characterized by a higher ratio of non-integrated versus integrated DNA)? Can the authors show examples of the DNA scope images ?

IFN-α/β is already known as a strong inducer of the cell surface activation marker, CD69, and thereby promoting lymphocyte retention in lymphoid organs via S1P1 (Shiow et al, Nature 2006). The fact that CD69 can be considered as an ISG might we worth to be added in the discussion. (Shiow et al, Nature 2006; doi: 10.1038/nature04606 ; Renson et al Vet Res 2010).

There is little discussion on the mechanisms of how IFN-I could mediate higher intracellular p24. It has indeed been shown that paradoxically, viral production can be enhanced by IFN-I (Seo et al, Science 2011). IFN-I regulates glucose metabolism (Burke et al, JVI 2014, doi: 10.1128/JVI.02649-13). HIV is expressed preferentially in CD4+ T cells with high glycolytic activity that are also characterized by high levels of ISGs (Valle Casuso et al, Cell Metabolism, DOI:https://doi.org/10.1016/j.cmet.2018.11.015). HIV-1, as many other viruses, thus exploits the metabolic environment for its replication cycle, and this might be a way to evade some restriction factors. Could it be that the pDC impacted the metabolic state of the CD4+ T cells through IFN-a? Would it be possible to analyse markers for higher metabolic/glycolytic activity in the co-cultured CD4+ T cells, such as Glut1 expression ? (Loisel-Meyer et al, PNAS 2012)

Line 403-405: while the increase of HLA-DR was less than for CD69, the HLA-DR increases on the CD4+ T cells after co-culture with PDC were still significant and consistently seen (Fig.S5C). It is thus not convincing to conclude that “the cells are maintained in a predominantly resting state” as some activation probably still have occurred. This does not change the overall conclusions of the study.

PLOS authors have the option to publish the peer review history of their article (what does this mean?). If published, this will include your full peer review and any attached files.

Reviewer #1: No

Reviewer #2: No
---

## [Decision Letter · Decision Letter 1]

1 Apr 2021

Dear Dr Nasr,

We are pleased to inform you that your manuscript 'Plasmacytoid dendritic cells have divergent effects on HIV infection of initial target cells and induce a pro-retention phenotype' has been provisionally accepted for publication in PLOS Pathogens.

Best regards,

Daniel C. Douek

Associate Editor

PLOS Pathogens

Richard Koup

Section Editor

PLOS Pathogens

Kasturi Haldar

Editor-in-Chief

PLOS Pathogens

orcid.org/0000-0001-5065-158X

Michael Malim

Editor-in-Chief

PLOS Pathogens

orcid.org/0000-0002-7699-2064

Reviewer Comments (if any, and for reference):

Reviewer's Responses to Questions

**Part I - Summary**

Reviewer #1: The reviewers adequately addressed all of my concerns.

Reviewer #2: The authors have sufficiently replied to the comments.

**Part II – Major Issues: Key Experiments Required for Acceptance**

Reviewer #1: None

Reviewer #2: (No Response)

**Part III – Minor Issues: Editorial and Data Presentation Modifications**

Reviewer #1: None

Reviewer #2: (No Response)

PLOS authors have the option to publish the peer review history of their article (what does this mean?). If published, this will include your full peer review and any attached files.

Reviewer #1: **Yes: **Mohamed Abdel-Mohsen

Reviewer #2: No

---

## [Editor Report · Acceptance letter]

9 Apr 2021

Dear Dr Nasr,

We are delighted to inform you that your manuscript, "Plasmacytoid dendritic cells have divergent effects on HIV infection of initial target cells and induce a pro-retention phenotype," has been formally accepted for publication in PLOS Pathogens.

Best regards,

Kasturi Haldar

Editor-in-Chief

PLOS Pathogens

orcid.org/0000-0001-5065-158X

Michael Malim

Editor-in-Chief

PLOS Pathogens

orcid.org/0000-0002-7699-2064